# NCX1 reverse mode promotes calcium-dependent Neutrophil Extracellular Trap formation and lung damage in chronic obstructive pulmonary disease

Shi-Xia Liao [1,2], Yan-Wen Wang[3], Ling-Mei Shi[2], Lan-Ying Zhang[2], Jian Zhou[2], Peng-Peng Sun[4], Huai-Yu Hu[5], Yu-Ting Liu[2], Xuan An[2], Jing-Qing Xu[6], Li Chen[7], Yao Ouyang [2,9] ✉, Yang Xu [1,9] ✉ & Ting-Hua Wang [1,7,8,9] ✉

Neutrophil-driven inflammation is central to the pathogenesis of chronic obstructive pulmonary disease (COPD). Emerging evidence suggests that $Ca^{2+}$ signaling is critical in regulating neutrophil activation, recruitment and tissue residency. In this study, we investigated the function of $Na^+/Ca^{2+}$ exchanger 1 (NCX1), a $Ca^{2+}$/cation membrane transporter, in neutrophils during COPD pathogenesis. Analysis of *human* specimens show that NCX1 is primarily upregulated in neutrophils from patients with mixed chronic bronchitis and emphysema. Cigarette smoke exposure induces NCX1 upregulation and promotes its reverse-mode transport activity, leading to elevated intracellular $Ca^{2+}$ levels and enhanced NETs formation. Neutrophil-specific genetic deletion of *Slc8a1* or pharmacological inhibition of NCX1 reverse transport effectively suppresses $Ca^{2+}$ influx, NETs release, and neutrophil accumulation and retention, thereby ameliorating chronic bronchitis and emphysematous changes. Collectively, our findings identify NCX1 as a regulator of $Ca^{2+}$-dependent NETs release in neutrophils. Targeting NCX1-mediated $Ca^{2+}$ influx or NETs formation represents a potential therapeutic strategy for neutrophilic inflammation in COPD.

Chronic obstructive pulmonary disease (COPD) is a progressive and debilitating lung disorder characterized by chronic bronchitis, fibrosis, small airway remodeling, and emphysema[1]. It is widely recognized that the development and progression of COPD are primarily driven by the inhaled noxious particles, particularly tobacco smoke and environmental pollutants, which promote activation and recruitment of neutrophils in the lungs, triggering persistent inflammatory responses[2,3]. Neutrophils are currently established as key players in the pathophysiology of COPD. Elevated sputum neutrophil counts have been shown to correlate with peripheral airway dysfunction, airway obstruction[4] and impaired lung function[5]. Clinical evidence suggests that therapeutic strategies targeting neutrophilic inflammation may

[1]Department of Anesthesiology, Department of Neurosurgery, State Key Laboratory of Biotherapy, West China Hospital, Sichuan university, Chengdu, China. [2]Department of Respiratory and Critical Care Medicine, Affiliated Hospital of Zunyi Medical University, Zunyi, China. [3]West China School of Medicine, West China Hospital, Sichuan University, Chengdu, China. [4]Department of Osteopathy, Affiliated Hospital of Zunyi Medical University, Zunyi, China. [5]ShenQi Ethnic Medicine College of Guizhou Medical University, Zunyi, China. [6]Department of Neurology, Affiliated Hospital of Zunyi Medical University, Zunyi, China. [7]Institute of Neurological Disease, West China Hospital, Sichuan University & The Research Units of West China, Chinese Academy of Medical Sciences, Chengdu, China. [8]Translational Neuromedicine Laboratory, Affiliated Hospital of Zunyi Medical University, Zunyi, China. [9]These authors jointly supervised this work: Ting-Hua Wang, Yang Xu, Yao Ouyang. ✉e-mail: ouyangyao@zmu.edu.cn; asdfxy@wchscu.cn; tinghua9988@wchscu.edu.cn

improve lung function, as indicated by increases in forced expiratory volume in one second (FEV1)[6]. However, neutrophil-targeted therapies remain challenging in COPD due to the risk of drug-induced neutropenia and subsequent infection[7]. Therefore, understanding the mechanisms underlying aberrant neutrophil activation and accumulation in the lungs is essential for developing safer and more effective treatments.

Disruption of intracellular $Ca^{2+}$ homeostasis in airway smooth muscle has been implicated in the pathogenesis of various lung diseases[8,9]. Airway neutrophilia is a hallmark feature of COPD, and further becomes pronounced during exacerbations[10]. Compelling evidence highlights the critical role of $Ca^{2+}$ signaling in regulating neutrophil functions, including NADPH oxidase activation, exocytosis, degranulation, and cytokine release[11–14]. Recent studies have also shown that $Ca^{2+}$ flux regulates neutrophil clustering and activation[15,16]. The $Na^+/Ca^{2+}$ exchangers (NCXs) represent a family of bidirectional membrane transporters that mediate the exchange of $Na^+$ and $Ca^{2+}$ in either direction, depending on the electrochemical gradient across the plasma membrane[17]. Elevation of $Na^+/Ca^{2+}$ exchanger 1 (NCX1, encoded by *Slc8a1*), in airway smooth muscle, has been shown to contribute to airway hyperresponsiveness in asthma[9]. Notably, data from *Human* Protein Atlas (https://www.proteinatlas.org/ENSG00000183023-Slc8a1/immune+cell) indicate that NCX1 (encoded by *Slc8a1*) is highly expressed in *human* neutrophils in contrast to other $Ca^{2+}$/cation membrane transporters and NCX family members. Nevertheless, little is known about the role of NCX1 in neutrophil function and COPD pathogenesis.

In the present study, we demonstrate that NCX1 is primarily expressed in *human* neutrophils, and cigarette smoke (CS) upregulates its expression, promoting the reverse mode of NCX1 transport and subsequently elevating intracellular $Ca^{2+}$ flux. Using a neutrophil-specific *Slc8a1* knockout *mouse* model, our work shows that NCX1 depletion or NCX1 reverse-mode inhibition ameliorates airway remodeling and preserves alveolar stem cells. This protective effect might be attributed to NCX1-mediated decline in $Ca^{2+}$ influx in neutrophils and reduced formation of neutrophil extracellular traps (NETs), which diminishes neutrophil accumulation during COPD and limits their retention during the stable phase, without impairing neutrophil differentiation, development, migration, or phagocytic activity. Our study identifies NCX1 as a regulator of $Ca^{2+}$-dependent NETs formation in COPD pathology. Pharmacological inhibition of NETs and NCX1 reverse transport offers a promising therapeutic strategy for COPD management.

## Results

### Increased neutrophilic NCX1 expression in patients with mixed chronic bronchitis and emphysema

*Human* lung tissues were obtained from Clinical Biobank of Sichuan University West China Hospital, with clinical characteristics and demographics detailed in Supplementary Table 1. In the lungs (encompassing distal airways and bronchioles, and adjacent alveolar structures) of patients with mixed chronic bronchitis and emphysema (Mixed CBE), NCX1 was predominantly localized to neutrophils (Fig. 1a), as corroborated by immune cell-specific expression profiles from the *Human* Protein Atlas database (https://www.proteinatlas.org/ENSG00000183023-Slc8a1/immune+cell). Notably, both mRNA and protein levels of NCX1 were significantly elevated in the lungs of patients with Mixed CBE compared to those with Controls or emphysema alone (Fig. 1b, c, Supplementary Fig. 1a, b). In parallel, the enhanced neutrophil infiltration and activity in the lung tissue of COPD patients was further validated by increased mRNA and protein levels of neutrophil activation markers, myeloperoxidase (MPO) and neutrophil elastase (NE) (Fig. 1d–h, Supplementary Fig. 1a, c, d). NCX1 mRNA levels were positively correlated with both MPO and NE expression levels (Fig. 1i, j), reinforcing its association with neutrophilic activation.

Immunofluorescence staining further confirmed the upregulation of NCX1 in Mixed CBE lung tissues, and its co-localization with CD66b$^+$ neutrophils (Fig. 1k). To validate the neutrophil-specific NCX1 enrichment, we performed flow cytometry with the bronchoalveolar lavage fluid (BALF) from Control and Mixed CBE patients, exhibiting prominently augmented neutrophils along with higher NCX1 mean fluorescence intensity in neutrophils of Mixed CBE patients (Fig. 1l–n). Across immune cell types, NCX1 expression was markedly enriched in BALF-derived neutrophils relative to other immune cell populations (macrophages, lymphocytes, etc.) in both Mixed CBE patients and controls (Supplementary Fig. 1e). Immunofluorescence of fluorescence-activated cell sorting (FACS)-sorted neutrophils further verified increased NCX1 expression in BALF-derived neutrophils from Mixed CBE patients (Fig. 1o). Importantly, this neutrophil-specific upregulation of NCX1 extended beyond the pulmonary compartment. Aligning with our lung tissue/BALF data, NCX1 protein levels were significantly elevated in peripheral blood neutrophils from Mixed CBE patients compared to controls (Supplementary Fig. 1f, g). These findings indicated NCX1 overexpression is a generalized neutrophil feature in COPD, detectable in both lung and circulatory compartments.

### NCX1 deletion mitigates neutrophils expansion and pathological changes of COPD

To functionally test the role of neutrophilic NCX1 in COPD pathogenesis, we next employed a CS-induced COPD *mouse* model. Following 12 weeks of CS exposure, mice developed hallmark pathological features, including emphysema and airway inflammation (Supplementary Fig. 2a, b). Consistent with *human* data, NCX1 was co-localized with neutrophils and significantly upregulated in lung tissues of these COPD mice, accompanied by elevated MPO and NE levels (Supplementary Fig. 2c–i). To determine whether neutrophil-specific NCX1 contributes causally to disease progression, we generated conditional knockout (cKO) mice using the Cre-loxP technology, specifically targeting *Slc8a1* deletion in neutrophils (*Slc8a1*$^{f+/f+}$;Mrp8-Cre$^+$), with genotyping confirmed by PCR (Supplementary Fig. 3a, b). Neutrophils were isolated from bone marrow, with purity verification by flow cytometry with 95%-98% (Supplementary Fig. 3c), and cell-type specificity of *Slc8a1* deletion was demonstrated by unaltered NCX1 expression in other immune populations (monocytes, macrophages, lymphocytes, and dendritic cells), while neutrophils from *Slc8a1* cKO mice exhibited nearly complete loss of NCX1 at both mRNA and protein levels (Supplementary Fig. 3d, e). Immunofluorescence further validated the effective ablation of NCX1 in neutrophils, showing an absent NCX1 signal in *Slc8a1* cKO mice relative to robust expression in control neutrophils (Supplementary Fig. 3f).

Both *Slc8a1*$^{f+/f+}$;Mrp8-Cre$^-$ and *Slc8a1*$^{f+/f+}$;Mrp8-Cre$^+$ mice were exposed to CS or air for 12 weeks (5 days/week), followed by pulmonary function tests prior to lung tissue collection post-euthanasia (Fig. 2a). Under air-exposed conditions, *Slc8a1* cKO and control mice showed comparable pulmonary function, indicating that basal respiratory mechanics were unaffected by NCX1 deletion. However, both genotypes showed significant impairment of lung function after CS exposure. Pressure–volume loop analysis revealed that CS exposure significantly decreased dynamic compliance (Cdyn) in both groups, while *Slc8a1* cKO mice retained higher Cdyn compared with CS-exposed controls (Fig. 2b, c). These results further support that neutrophil-specific NCX1 deficiency partially preserves lung compliance and mitigates airway resistance under chronic CS exposure. Consistently, significant airflow limitation (decreased FEV$_{20(ms)}$/FVC and FEV$_{50(ms)}$/FVC) and pulmonary hyperinflation (increased FRC/TLC) were shown in CS-exposed control mice (Fig. 2d, e, Supplementary Fig. 4a. However, *Slc8a1* cKO mice were partially protected from lung function decline relative to CS-exposed controls, improving airflow limitation (higher FEV$_{20(ms)}$/FVC and FEV$_{50(ms)}$/FVC ratios) and reducing hyperinflation (lower FRC/TLC) (Fig. 2d, e,

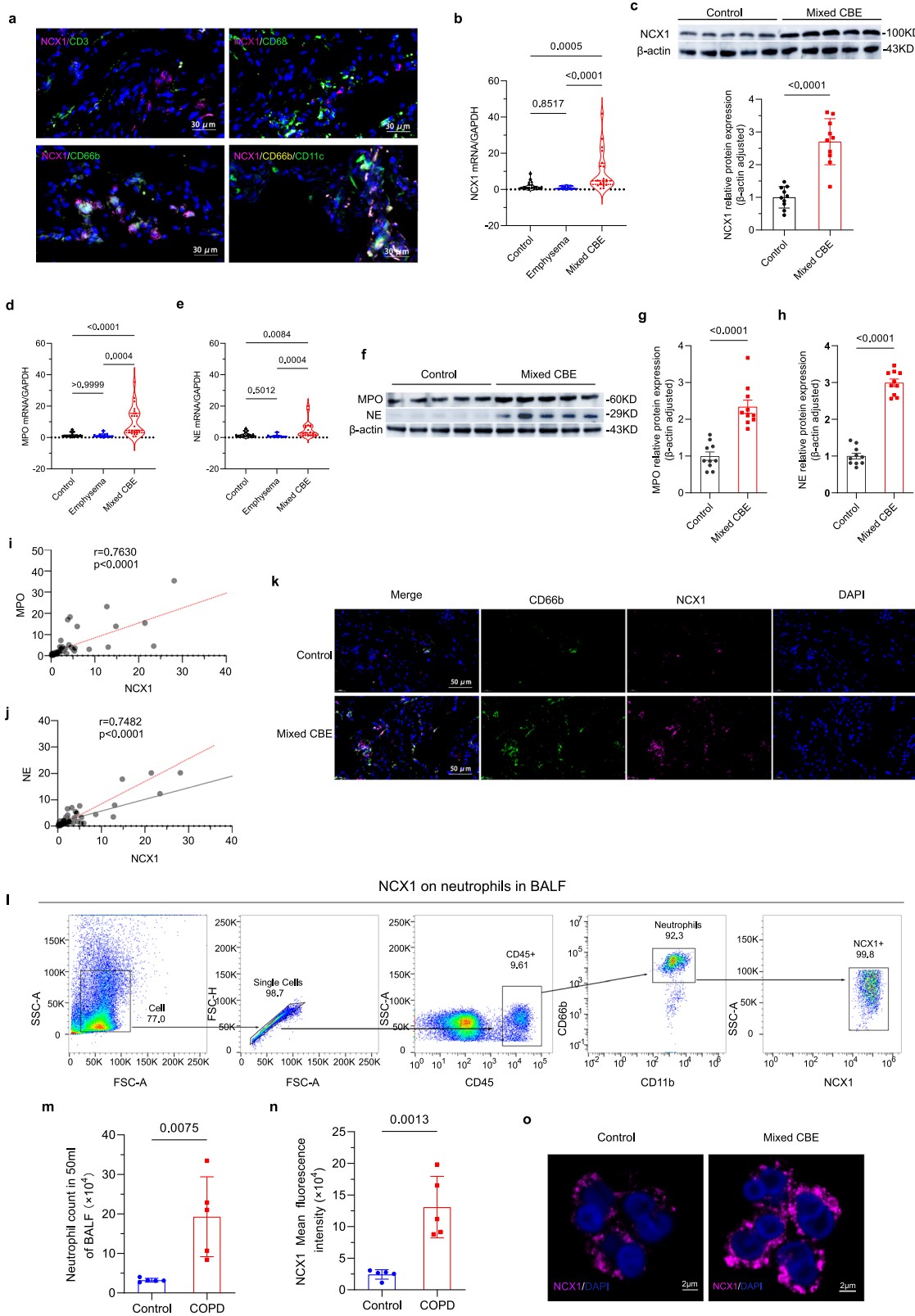

Supplementary Fig. 4a). Meanwhile, histological and cytometric analyses revealed that under baseline conditions, both genotypes had similar neutrophil infiltration, lung morphology (MLI), collagen deposition area, airway smooth muscle thickness, and alveolar stem cells numbers (SFTPC⁺). In contrast, following CS exposure for 12 weeks, *Slc8a1* cKO mice exhibited significant protection against CS-triggered neutrophilic inflammation, airspace enlargement,

collagen deposition expansion, airway remodeling, and loss of alveolar stem cells (Fig. 2f–o). These results demonstrate that neutrophil-specific deletion of NCX1 confers significant protection against CS-induced COPD-characteristic airway dysfunctions and key pathological features, without adversely affecting normal lung structure and function.

**Fig. 1 | The expression of NCX1 in neutrophils increased in lung tissue of COPD patients. a** Representative immunofluorescence images of NCX1 (purple) co-localization with different immune cell-type markers (CD3⁺, T cells, CD68⁺, mac-rophages, CD66b⁺, neutrophils; CD11c⁺, neutrophils/dendritic cells) in distal air-ways and bronchioles and adjacent alveolar regions from Mixed CBE patients ($n = 5$ patients). Blue, DAPI-stained nuclei. Light blue indicates colocalization of NCX1 with neutrophils. Scale bar = 30 μm. **b** RT-qPCR analyzes mRNA levels of NCX1 in *human* lung tissue from Control patients ($n = 20$), emphysema patients ($n = 10$), and Mixed CBE patients ($n = 20$). **c** Western blot analysis of NCX1 protein levels in lung tissues from Control patients and Mixed CBE patients ($n = 10$), normalized to β-actin and displayed relative to controls. Uncropped blots in Source Data. **d, e** RT-qPCR analyzes mRNA levels of MPO and NE in lung tissues from Control individuals ($n = 20$), emphysema patients ($n = 10$), and Mixed CBE patients ($n = 20$). **f–h** Wes-tern blot analysis of MPO and NE expression in lung tissues from Control patients and Mixed CBE patients ($n = 10$), normalized to β-actin and displayed relative to controls. Uncropped blots in Source Data. **i, j** Spearman's correlation analyzes the correlation of NCX1 expression with MPO and NE levels. 95% CI: 0.5281 to 0.8724 for (**i**), 0.7029 to 1.007 for (**j**). **k** Representative immunofluorescence images show

expression and localization of CD66b (green) and NCX1 (purple) in lung tissues of Control and Mixed CBE patients. Scale bar = 50 μm. Blue, DAPI-stained nuclei. **l–n** Flow cytometry identifies neutrophil numbers (**m**) and NCX1 mean fluorescence intensity (**n**) on neutrophils in *human* BALF ($n = 5$ individuals). **o** Representative immunofluorescence images show expression of NCX1 in neutrophils from *human* BALF of Control and Mixed CBE patients. Scale bar = 2 μm. Blue, DAPI-stained nuclei. Data point represents one biologically independent replicate with three technical replicates (**b, d, e**), one biologically independent replicate with two technical replicates (**c, g, h**). Data points represent biologically independent replicates (**m, n**). Quantitative data are presented as min to max with all points mean (**b, d, e**), and as Mean ± SD (**c, g, h, m, n**). Two-sided *t*-test (**g, h, m, n**) and one-way ANOVA with Tukey's multiple comparison test (**b, d, e**) were used to calculate the *p*-values. At least 3 times, each experiment was independently repeated with similar results. Source data are provided as a Source Data file. CBE chronic bron-chitis and emphysema; MPO myeloperoxidase; NE neutrophil elastase; BALF bronchoalveolar lavage fluid; SCC-A forward scatter-area; FSC-H forward scatter-height; FSC-A side scatter-area; ns no significance.

## *Slc8a1* deletion impaired NETs formation without affecting neutrophil development, maturation, migration and phagocytosis

To investigate the contribution of NCX1 in neutrophil function, we conducted a series of in vitro assays with CD45⁺CD11b⁺Ly6g⁺ neu-trophils purified from bone marrow of non-CS-exposed *Slc8a1*^f+/f+;Mrp8-Cre⁻ mice and *Slc8a1*^f+/f+;Mrp8-Cre⁺ mice using FACS. Flow cytometry, Diff-quick staining and scanning electron microscopy revealed no obvious morphological differences between the two genotypes (Fig. 3a–c). Genetic deletion of *Slc8a1* in neutrophils did not alter neutrophilic development and maturation (Fig. 3d–f). Moreover, total neutrophil counts in the bone marrow, peripheral blood, and lungs, along with white blood cell count in peripheral blood, remained comparable between cKO and control groups (Fig. 3g–j), suggesting that *Slc8a1* deficiency did not impair neutrophil migration ability. Phagocytic function was also preserved using neutrophils sorted from bone marrow, as shown by immunohistochemical staining and quan-tified using pHrodo Red-conjugated Escherichia coli particles in flow cytometry assays, with no significant differences in phagocytic effi-ciency or index (Fig. 4a–f).

We next assessed the role of NCX1 in NETs formation. Following 4-hour PMA stimulation at 37 °C, bone marrow neutrophils from *Slc8a1*^f+/f+;Mrp8-Cre⁻ neutrophils produced obvious NETs, which was decreased in *Slc8a1*^f+/f+;Mrp8-Cre⁺ mice (Supplementary Fig. 4b). Similarly, bone marrow-derived neutrophils from *Slc8a1* cKO mice exhibited markedly impaired NETs formation as evidenced by reduced NETs area and diminished levels of Cit-H3 (Citrullinated Histone H3) (Fig. 5a–c), indicating a specific role for NCX1 in neutrophil patho-genicity. Consistently, *human* lung specimens from Mixed CBE patients showed enhanced NETs compared to Controls (Fig. 5d, e). Notably, NETs were significantly reduced in *Slc8a1*^f+/f+;Mrp8-Cre⁺ mice following CS exposure for 12 weeks, with decreased NET area and Cit-H3 expression (Fig. 5f–h). Whereas no differences of Cit-H3 levels were observed under air-exposed conditions (Fig. 5h).

To investigate whether the persistence of neutrophils in the lung after CS exposure cessation correlates with NETs formation, we assessed the abundance of tissue-resident neutrophils (TRNs) in lungs of *Slc8a1*^f+/f+;Mrp8-Cre⁻ mice and *Slc8a1*^f+/f+;Mrp8-Cre⁺ mice two weeks after cessation of CS (Fig. 5i). To distinguish between circulating and TRNs, we performed intravenous (i.v.) injection of Alexa Fluor 700-conjugated anti-CD45 labeling prior to the lung harvest enabled dis-tinction between circulating (Alexa Fluor anti-CD45⁺ cells) and resident neutrophils (anti-APC-Cy7 CD45⁺/CD11b⁺F4/80-Gr1⁺Ly6c-Ly6g⁺, refer-red to as TRNs in murine lungs) (Fig. 5j). Flow cytometry analyses confirmed that *Slc8a1* deletion in neutrophils significantly reduced the number of TRNs post-CS exposure (Fig. 5k). Our findings

demonstrated that the accumulation of TRNs in the COPD lungs is closely linked to NCX1 which impeded the normal regression of neu-trophils from the lung. We further explored whether NCX1 sustains a pro-inflammatory microenvironment that favors neutrophil retention. Indeed, CS exposure triggered significant upregulation of key pro-inflammatory mediators (TNF, CXCL-2, IL-6, IL-1β, and IL-17a)[18,19] in Control mice, whereas their levels were markedly suppressed in *Slc8a1* cKO mice (Fig. 5l–p), highlighting that NCX1 sustains neutrophil retention by creating a chemokine-rich microenvironment. Given prior studies elucidating that NETs stimulate chemokine production in lung epithelium[20] and fibroblasts[21], we speculate that NCX1-driven NET formation might be a key upstream regulator of chemokine-rich microenvironment, favoring neutrophil retention.

## NETs blockade depressed the neutrophils expansion and NETs formation to attenuate lung parenchymal cell loss and airway remodeling

To determine whether CS-induced NETs formation drives the pathological changes in COPD, we pharmacologically inhibited NETs formation using Cl-amidine in mice every two days during the course of CS exposure (Fig. 6a). Cl-amidine is used as a pan-peptidylarginine deiminase (PAD) inhibitor to broadly suppress NETs formation, given its demonstrated efficacy in reducing citrullination-dependent NETs release[22]. Pulmonary function tests revealed a marked reduction of dynamic compliance after CS exposure, indicated by pressure–volume loop data, which was alleviated by Cl-amidine treatment (Fig. 6b, c). Meanwhile, CS exposure significantly impaired respiratory mechanics compared to Control mice, manifesting as decreased FEV$_{20(ms)}$/FVC and FEV$_{50(ms)}$/FVC ratios, along with increased FRC/TLC (Fig. 6d, e, Supplementary Fig. 4c), indicative of airflow limitation and pulmonary hyperinflation. Remarkably, Cl-amidine administration substantially attenuated these CS-induced functional damage, with FEV$_{20(ms)}$/FVC and FEV$_{50(ms)}$/FVC ratios showing significant recovery and FRC/TLC returning toward baseline levels (Fig. 6d, e, Supplementary Fig. 4c). These improvements demonstrate that NETs inhibition depressed airflow limitation and increased residual volume in COPD pathogenesis. After 12 weeks of CS exposure, ELIZA confirmed elevated Cit-H3 levels in *mouse* BALF, and flow cytometry showed a marked expansion of pulmonary neu-trophils compared to the Control group, while immunofluorescence staining revealed increased NETs area in lung tissues (Fig. 6f–i). Treatment with Cl-amidine significantly decreased neutrophil accu-mulation, reduced Cit-H3 levels and NETs area compared to the CS+Vehicle and CS group (Fig. 6f–i), confirming its efficacy in sup-pressing NETs release in vivo. H&E staining displayed that CS expo-sure induced severe emphysematous destruction and airway

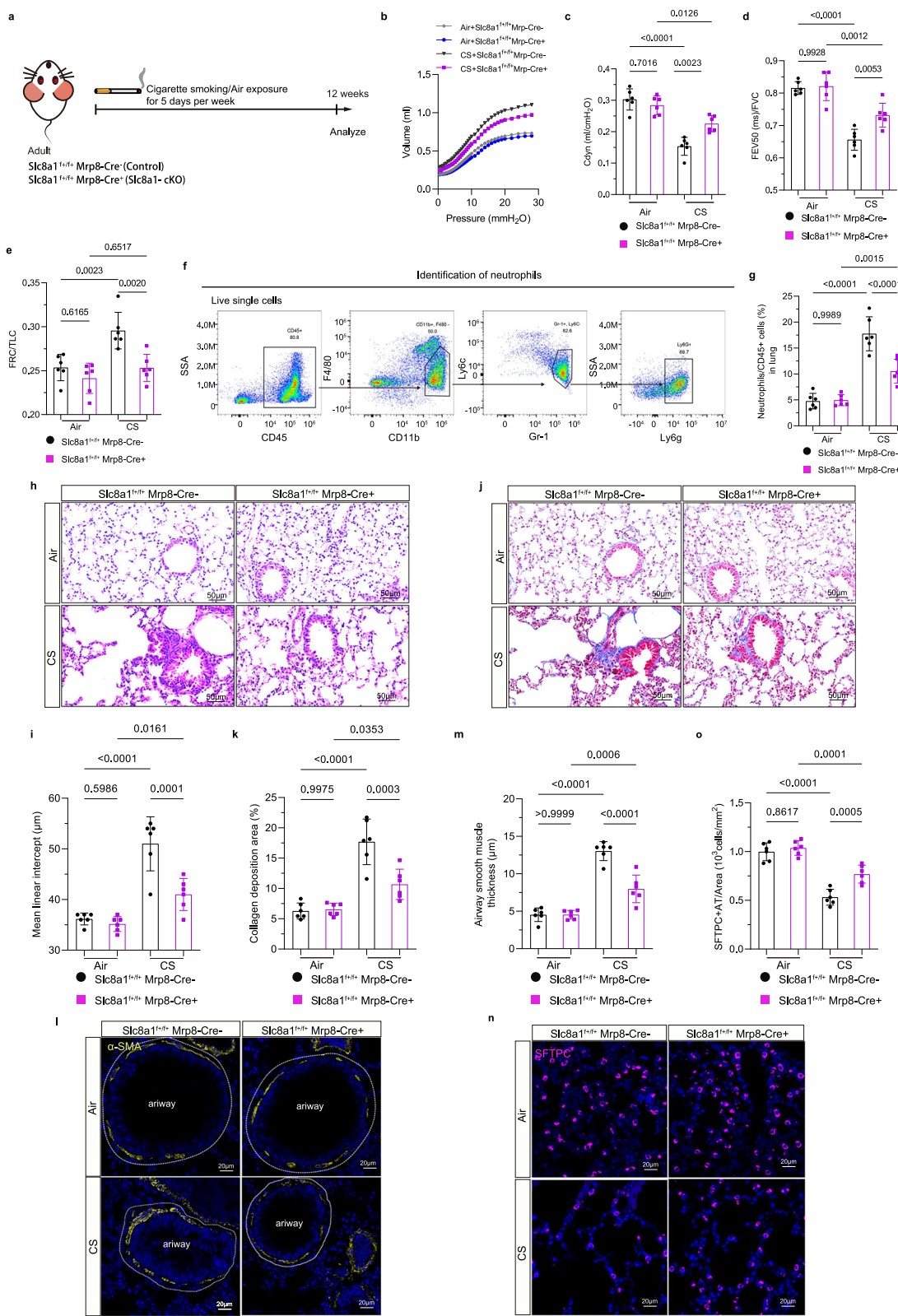

inflammation, which was notably attenuated by Cl-amidine treatment (Fig. 6j, k). Furthermore, Masson's Trichrome staining and immunofluorescence for a-SMA and SFTPC revealed that Cl-amidine effectively suppressed airway remodeling and preserved alveolar epithelial progenitor cells. Specifically, CS exposure triggered pronounced peribronchial fibrosis, thickened airway smooth muscle, and depleted alveolar stem cells, whereas Cl-amidine administration markedly reduced collagen deposition, diminished bronchial smooth muscle hyperplasia, and restored alveolar stem cells (Fig. 6l–q). These findings suggested that inhibiting NET formation reversed CS-induced pulmonary structural and functional damage, linking NET inhibition to alveolar preservation, airway remodeling suppression, and small airway function improvement.

**Fig. 2 | *Slc8a1* deficiency in neutrophils mitigated the pathological changes of COPD mice. a** Schematic diagram of CS exposure protocol in *Slc8a1*$^{f+/f+}$;Mrp8-Cre$^+$ (*Slc8a1* cKO mice) and littermate *Slc8a1*$^{f+/f+}$;Mrp8-Cre$^-$ (Control) mice. COPD is induced in *Slc8a1*$^{f+/f+}$;Mrp8-Cre$^+$ and *Slc8a1*$^{f+/f+}$;Mrp8-Cre$^-$ mice via CS exposure for 12 weeks before lung tissue collection. *Slc8a1*$^{f+/f+}$;Mrp8-Cre$^+$ and *Slc8a1*$^{f+/f+}$;Mrp8-Cre$^-$ mice were exposed under normal air in the Control group. **b** The pressure-volume curves were generated in the mice, presenting the mean values of volume measured under different airway pressures. **c–e** Pulmonary function parameters, including dynamic compliance data, FEV$_{50(ms)}$/FVC and FRC/TLC, were measured after 12-week CS exposure (*n* = 6 mice). **f** Flow cytometry strategy identifies neutrophils from the lung tissues of *Slc8a1*$^{f+/f+}$;Mrp8-Cre$^+$ and *Slc8a1*$^{f+/f+}$;Mrp8-Cre$^-$ mice under air or CS exposure. **g** Flow cytometry quantification of neutrophil proportions in lung tissues (*n* = 6 mice). **h, i** Representative H&E-stained lung sections showing alveolar destruction and airway inflammation, and corresponding mean linear intercept values (*n* = 6 mice). Scale bar = 50 μm. **j, k** Masson's Trichrome staining and quantification of lung collagen deposition area (*n* = 6 mice). Scale bar = 50 μm. **l–o** Immunofluorescence and quantification for α-SMA (yellow) and SFTPC (purple) showing airway remodeling, smooth muscle hyperplasia, and loss of alveolar epithelial stem cells (*n* = 6 mice). Scale bar = 20 μm. Each data point represents one biologically independent replicate with three technical replicates (**b–e, i, k, m, o**), and one biologically independent replicate (**g**). All quantitative data are presented as Mean ± SD. One-way ANOVA with Tukey's multiple comparison test was used to calculate the *p*-values for all quantitative data in this figure. At least 3 times, each experiment was independently repeated with similar results. Source data are provided as a Source Data file. CS cigarette smoke; cKO conditional knockout; Cdyn dynamic compliance; FEV$_{50(ms)}$ forced expiratory volume in 50 ms; FVC forced expiratory volume; FRC functional residual capacity; TLC total lung capacity; ns no significance.

## Inhibition of NCX1 reverse mode activity decreased Ca$^{2+}$ flux and NETs release

Multiple neutrophil-activating stimuli, including phorbol 12-myristate 13-acetate (PMA), N-formyl-methionyl-leucyl-phenylalanine (fMLP) and cigarette smoke extract (CSE), were found to upregulate the expression of NCX1, with CSE showing the most robust effect, as confirmed by RT-qPCR, WB, and immunofluorescence staining (Fig. 7a–d). Importantly, NCX1 functions bidirectionally, operating in both forward and reverse transport modes. To elucidate the underlying mechanism by which NCX1 enhances NETs release, we examined the transport patterns of NCX1 after CSE exposure. In vitro Ca$^{2+}$ flux detection showed that CSE induced a sharp increase in Ca$^{2+}$ levels within neutrophils, but this Ca$^{2+}$ influx was effectively suppressed by pretreatment with KB-R7943, an inhibitor of NCX1 reverse transport (Fig. 7e). This finding may suggest that CSE induced Ca$^{2+}$ overload in neutrophils primarily through reverse transport of NCX1. Najder et al. reported that the increase of intracellular Na$^+$ concentration can reverse the activity of NCX1 in neutrophils[23]. To dissect whether CSE acts directly on NCX1 or indirectly via Na$^+$ dynamics, we simultaneously measured real-time changes in intracellular Na$^+$ ([Na$^+$]i) and Ca$^{2+}$ ([Ca$^{2+}$]i) in neutrophils stimulated with three agents (CSE, PMA, and fMLP) by ion-sensitive fluorescent probes. Consistent with the study of Najder et al., fMLP induced concurrent increases in both [Na$^+$]i and [Ca$^{2+}$]i, indicating indirect NCX1 reverse-mode activation secondary to Na$^+$ rise (Fig. 7f, g). In contrast, CSE and PMA triggered a rapid [Na$^+$]i decrease alongside a robust [Ca$^{2+}$]i increase (Fig. 7f, g). This divergence demonstrated that CSE and PMA might act via a distinct, direct mechanism favoring reverse-mode NCX1 operation (Ca$^{2+}$ influx/Na$^+$ efflux). Moreover, CSE-induced [Ca$^{2+}$]i rise was nearly abolished in NCX1-deficient neutrophils (Fig. 7h), proving NCX1 is the primary route for CSE-triggered Ca$^{2+}$ influx.

Considering the possibility that mechanical stimulation during drug application might transiently activate mechanosensitive ion channels such as Piezo, we simultaneously pretreated neutrophils with GsMTx4, a specific inhibitor of Piezo channels[24], to rule out this potential artifacts. Inhibition of Piezo channels delayed the time to reach peak Ca$^{2+}$ and Na$^+$ fluxes compared to untreated cells, yet the magnitude and stability of the sustained fluxes remained unchanged (Supplementary Fig. 4d, e). Similarly, GsMTx4 pretreatment did not alter the trend of CSE-stimulated Ca$^{2+}$ flow in mice of different genotypes, indicating that the observed Ca$^{2+}$ and Na$^+$ fluxes primarily reflect true NCX1-dependent transport rather than mechanical artifacts. Furthermore, to further confirm that NCX1 is directly involved, we overexpressed NCX1 in CHO-K1 cells and observed the effects on intracellular ion dynamics (Supplementary Fig. 4g–k). CSE stimulation of NCX1-overexpressing CHO-K1 cells led to a pronounced increase in [Ca$^{2+}$]i and a significant decrease in [Na$^+$]i compared to control cells (Supplementary Fig. 4j, k). These results provide further evidence that enhanced NCX1 expression directly facilitates reverse-

mode activity (Ca$^{2+}$ influx coupled with Na$^+$ efflux) during CSE exposure.

Consistently, NCX1 depletion significantly impaired CSE-induced NETs formation in neutrophils from *Slc8a1*$^{f+/f+}$;Mrp8-Cre$^-$ and *Slc8a1*$^{f+/f+}$;Mrp8-Cre$^+$ mice, as shown by reduced Cit-H3 levels and NETs area (Fig. 7i, j). Based on these results, we hypothesized that CSE triggers a reverse transport pattern of NXC1, which increases Ca$^{2+}$ inflow and subsequent NETs release, ultimately exacerbating COPD. To further validate this mechanism, we conducted in vitro experiments and observed that treatment with the NCX1 reverse-mode inhibitor KB-R7943 decreased NETs formation and Cit-H3 levels in CSE- or PMA-stimulated neutrophils (Fig. 7k, l, Supplementary Fig. 4l, m). We further assessed the functional association of NCX1 with Ca$^{2+}$ influx, and found that the inhibitory effect of KB-R7943 administration on CSE-induced Ca$^{2+}$ inflow was abolished in neutrophils from *Slc8a1*$^{f+/f+}$;Mrp8-Cre$^+$ mice (Supplementary Fig. 4n), supporting the specificity of KB-R7943 and crucial role of NCX1 in neutrophil Ca$^{2+}$ dynamics. Pre-treatment with BAPTA-AM, a cytosolic Ca$^{2+}$ chelator, significantly depressed CSE-triggered [Ca$^{2+}$]i influx, Cit-H3 expression, and NETs release (Fig. 7k, l, Supplementary Fig. 4o, p). Conversely, treatment with Ca$^{2+}$ ionophore A-23187 elevated [Ca$^{2+}$]i, robustly inducing Cit-H3 expression and NETs formation (Fig. 7k, l; Supplementary Fig. 4o, p), confirming Ca$^{2+}$ overload alone is sufficient to drive NETs formation. These findings underscore the critical role of NCX1-mediated Ca$^{2+}$ influx in driving NETs formation and COPD pathogenesis.

## NCX1 reverse mode inhibition preserved pulmonary function and mitigated CS-induced injury

To evaluate the therapeutic potential of targeting NCX1 reverse-mode activity in COPD, we administered KB-R7943 to mice subjected to CS exposure (Fig. 8a). Under normal air-exposed conditions, pressure–volume loop analysis demonstrated that CS exposure significantly reduced dynamic compliance (Cdyn), while KB-R7943 treatment largely restored Cdyn toward normal levels (Fig. 8b, c). These results provide functional evidence that pharmacological inhibition of NCX1 reverse-mode activity mitigates CS-induced decline in lung compliance. Additional pulmonary function tests revealed that KB-R7943 treatment alone does not alter baseline lung function, as Control and KB-R7943-alone groups exhibited no significant differences in FEV$_{20(ms)}$/FVC, FEV$_{50(ms)}$/FVC, FRC/TLC data (Fig. 8d, e, Supplementary Fig. 4q). In contrast, CS-exposed mice showed severe functional impairment, including significant airflow limitation and pulmonary hyperinflation, which were significantly attenuated by KB-R7943 treatment (Fig. 8d, e). The protective effects of KB-R7943 treatment were evidenced by a substantial reduction in NETs formation, indicated by decreased protein levels of Cit-H3 and shrunk NETs area compared to the CS group (Fig. 8f–h). No obvious differences were observed in NETs release and area between Control and KB-R7943 treatment groups under air exposure (Fig. 8f–h), supporting the

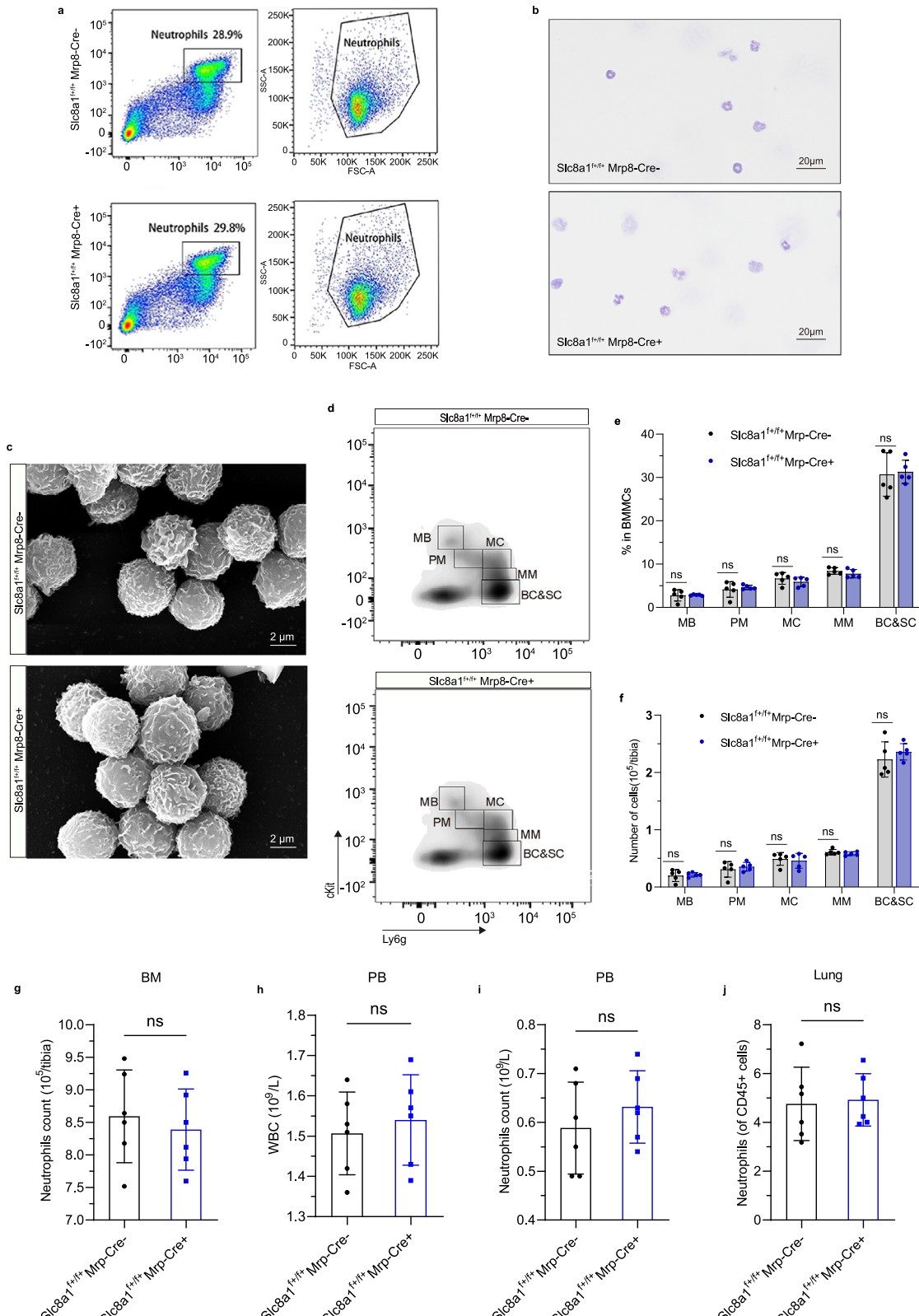

specificity and safety of the inhibitor under physiological conditions. Furthermore, CS exposure-induced emphysematous alveolar destruction, peribronchial fibrosis, airway smooth muscle hyperplasia, and depletion of alveolar epithelial progenitor cells were reversed with KB-R7943 treatment in CS-exposed mice. Specifically, H&E staining demonstrated that KB-R7943 preserved alveolar architecture (Fig. 8i, j), while Masson's trichrome staining and immunofluorescence for α-

SMA and SFTPC confirmed mitigation of fibrotic remodeling and restoration of alveolar stem cells (Fig. 8k–p). No significant alterations were shown between the Control and KB-R7943 treatment groups under air exposure (Fig. 8k–p). These findings collectively demonstrated that pharmacological inhibition of NCX1 reverse mode activity protected against CS-induced COPD pathogenesis without affecting baseline pulmonary function.

**Fig. 3 | Genetic deletion of neutrophil *Slc8a1* does not alter neutrophil development, maturity and migration. a** Flow cytometry analysis of bone marrow neutrophil proportions. **b, c** The morphology of bone marrow-derived neutrophils was visualized by Diff-Quik staining and scanning electron microscope. Scale bar = 20 μm (**b**), 2 μm (**c**). **d−f** The development and maturation of bone marrow neutrophils, as well as the proportions of each subpopulation (*n* = 5 mice). **g** The number of bone marrow neutrophils in the tibia of mice with different genotypes (*n* = 6 mice). **h, i** The number of white blood cells and neutrophils in peripheral blood (*n* = 6 mice). **j** The proportion of neutrophils to CD45⁺ cells in lung tissues

(*n* = 6 mice). Each data point represents one biologically independent replicate (**e−j**). Neutrophils were derived from the bone marrow, blood, and lung tissues of mice with non-CS exposure. All quantitative data shown here are presented as Mean ± SD. A two-sided *t*-test was used to calculate the *p*-values for all quantitative data. At least 3 times, each experiment was independently repeated with similar results. Source data are provided as a Source Data file. MB myeloblast; PM promyelocyte; MC myelocyte; MM metamyelocyte; BC band cell; SC segmented neutrophil; BM bone marrow; PB peripheral blood; PMNs polymorphonuclear neutrophils; ns no significance.

## Discussion

COPD is a challenging condition characterized by high morbidity and mortality, with neutrophil-driven inflammation being a prevalent phenotype[2,25]. Despite this, effective therapeutic strategies targeting neutrophils in COPD remain a significant unmet clinical need[6,26]. Through integrative analysis of *human* lung tissues, bronchoalveolar lavage fluid, and peripheral blood, we demonstrate that NCX1 expression is primarily upregulated in neutrophils from patients with COPD. This enrichment of NCX1 in pulmonary exacerbated neutrophilic activation and NETs formation, thereby contributing to neutrophil accumulation and retention in the lungs. Importantly, genetic ablation of *Slc8a1* or pharmacological blockade of NCX1 reverse-mode activity in neutrophils conferred marked protection against airway dysfunctions and hallmark pathological features through diminished cellular Ca²⁺ influx and NETs formation. Our results provide mechanistic clarity, linking CS-induced NETs formation to NCX1-mediated calcium influx, not only supporting a causal role for NCX1 in orchestrating NETs-driven COPD injury, but also implicating its potential as a viable therapeutic target in COPD.

In COPD, the most important pathological changes including airway remodeling changes (airway wall fibrosis and bronchial smooth muscle hypertrophy), and emphysema changes (characterized by loss of alveolar epithelium), are driven by the accumulation of inflammatory exudates in the small airways and lung tissues[27−29]. Here, our data firstly demonstrated that NCX1, encoded by *Slc8a1*, was predominately elevated in the lung tissues, BALF, and peripheral blood of patients with COPD, and primarily expressed in neutrophils relative to other immune cell types. This systemic pattern of NCX1 elevation supports the hypothesis that NCX1-driven NETs formation may contribute to extrapulmonary COPD comorbidities. NCX1 expression was correlated positively with expression of neutrophil activation markers MPO and NE, suggest that NCX1 may play a crucial role in COPD pathogenesis by modulating inflammatory responses and tissue remodeling. Previous studies have primarily focused on the role of NCX1 in cardiac and neuronal tissues, where it regulates intracellular calcium homeostasis and contributes to excitability and injury responses under stress conditions[17,30,31]. Its involvement in chronic pulmonary diseases has been further explored in this study by employing neutrophil-specific *Slc8a1* deletion in mice, which confirmed the striking function of neutrophilic NCX1 in CS-induced COPD pathogenesis. Genetic ablation of *Slc8a1* significantly attenuated hallmark features of COPD, including airflow limitation, emphysema, airway remodeling, and loss of alveolar stem cells, accompanied by a marked reduction in CS-induced pulmonary neutrophil accumulation. Absent impact of *Slc8a1* ablation on pulmonary functions, lung morphology and neutrophil amount were revealed in mice under normal air exposure. This distinguishes our work from prior studies in which modulation of neutrophilic inflammation often led to systemic immunosuppression or impaired host defense[32,33]. Blockade of CXCR2, a key chemokine receptor mediating neutrophil recruitment, has shown anti-inflammatory benefits in COPD models but was associated with increased risk of infections in clinical trials[34]. Unlike such broad immunosuppressive strategies, targeting NCX1 appears to selectively dampen pathogenic neutrophilic responses in the lung while preserving essential immune functions. This

specificity may be linked to NCX1's role in regulating Ca²⁺ influx in activated immune cells, particularly under inflammatory microenvironments, as previously suggested in macrophage and monocytes studies[35].

To further clarify the functional role of NCX1 in neutrophils, we systemically investigated multiple key aspects of neutrophil biology, including morphology, development, migration, and phagocytosis[15,36−38]. Notably, NCX1 deletion in neutrophils did not affect these baseline properties, suggesting that NCX1 is dispensable for neutrophil development and basic immune functions under physiological conditions. However, when examining NETs formation, a specialized function critically in both host defense and chronic inflammation[39], we observed a prominent increase of NETs formation in *human* lungs and *mouse* lungs in the context of COPD. Striking reduction in NETs release was reversely revealed in PMA-induced bone marrow neutrophils from *Slc8a1* cKO mice, along with NETs reduction in lung tissues of *Slc8a1* cKO mice. NETs are web-like extrusions composed of DNA and anti-microbial proteins (MPO, NE, proteinase 3, cathepsin G, etc), with initially recognized for their pathogen-killing abilities, but accumulating evidence highlights their detrimental role in chronic lung diseases, particularly through the amplification of tissue injury and inflammation[40−43]. Previous reports identified abundant NETs in the lungs of COPD patients[20,44], and indicated that NETs can damage epithelial and stromal cells, thereby impairing tissue repair[21,45,46]. Our findings revealed that pharmacological inhibition of NETs formation by Cl-amidine in mice not only attenuated pulmonary dysfunctions, CS-induced neutrophil accumulation, collagen deposition and smooth muscle thickening around the airway, but also attenuated the loss of alveolar stem cells−key components for lung regeneration. Our study provides insights into the mechanistic contribution of NETs to COPD pathogenesis. We hypothesized that elevated NETs release drives persistent neutrophil recruitment and retention in lungs, thereby leading to airway remodeling and emphysema in COPD. Even during stable (non-exacerbation) phases of COPD, we found large numbers of neutrophils persisting in the airways, representing as a hallmark of airway inflammation in stable COPD. TRNs, a crucial pathogenic neutrophil subset, have been recognized as key players implicated in COPD pathology, promoting chronic inflammation, sustaining tissue damage, and accelerating disease progression[44]. We investigated whether NCX1 plays a crucial role in neutrophil retention within the lungs during non-exacerbation phases of COPD by establishing a CS-induced COPD model using mice of different genotypes. After 12 weeks of CS exposure, mice were transferred to a smoke-free environment for an additional 2 weeks to assess the resolution of lung-infiltrating neutrophils. Our finding revealed that specific deletion of NCX1 in neutrophils significantly reduced the number of TRNs in lung tissues, suggesting that NCX1 is critically involved in neutrophil retention within the lungs, and that NETs may be a key factor mediating this process during stable COPD. In line with prior work elucidating that NETs stimulate chemokine production in lung epithelium[20] and fibroblasts[21], we also observed that CS exposure significantly increased key pro-inflammatory mediators, whereas these cytokines and chemokines were markedly suppressed in *Slc8a1* cKO lungs. Our data suggest that NCX1 sustains a pro-inflammatory

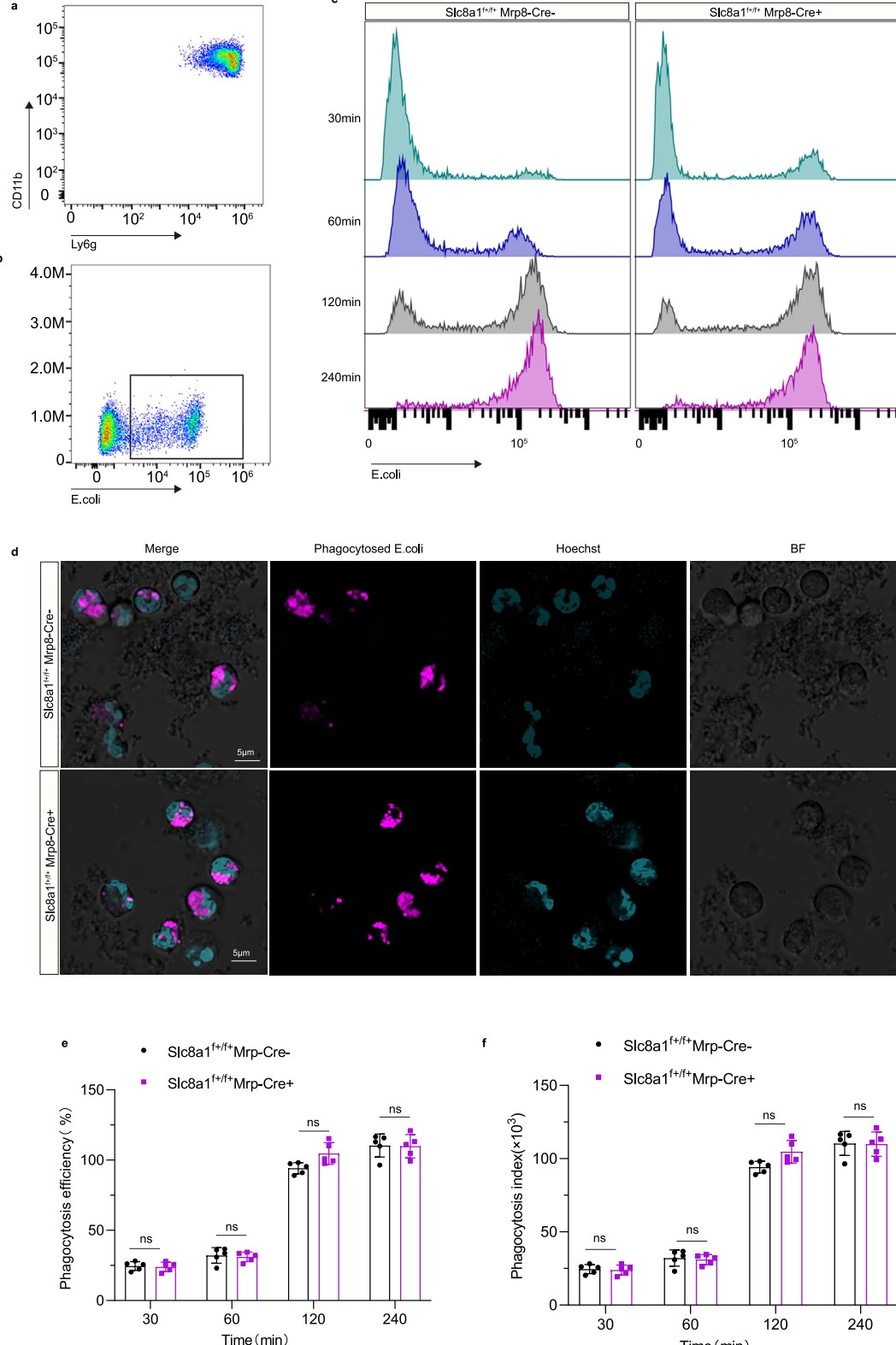

**Fig. 4 | *Slc8a1* deletion does not affect the phagocytosis capability of neutrophils. a** Flow cytometry analysis shows the purity of neutrophils isolated from the bone marrow of *Slc8a1*[f+/f+];Mrp8-Cre⁻ and *Slc8a1*[f+/f+];Mrp8-Cre⁺ mice. **b, c** Flow cytometry analysis shows the phagocytosis of neutrophils to E. coli bioparticles at different time points. **d** In vitro phagocytosis assay. Purified neutrophils were stimulated with pHrodo Red-conjugated Escherichia coli bioparticles for 2 h and photographed by confocal microscopy. Scale bar = 5 μm. **e, f** Quantification of phagocytosis efficiency and index between two genotype mice (*n* = 5) at 30, 60, 120 and 240 min. Each data point represents one biologically independent replicate (**e, f**). All quantitative data are presented as Mean ± SD. A two-sided *t*-test was used to calculate the *p*-values for all quantitative data. At least 3 times, each experiment was independently repeated with similar results. Source data are provided as a Source Data file. BF bright field; min minutes; ns no significance.

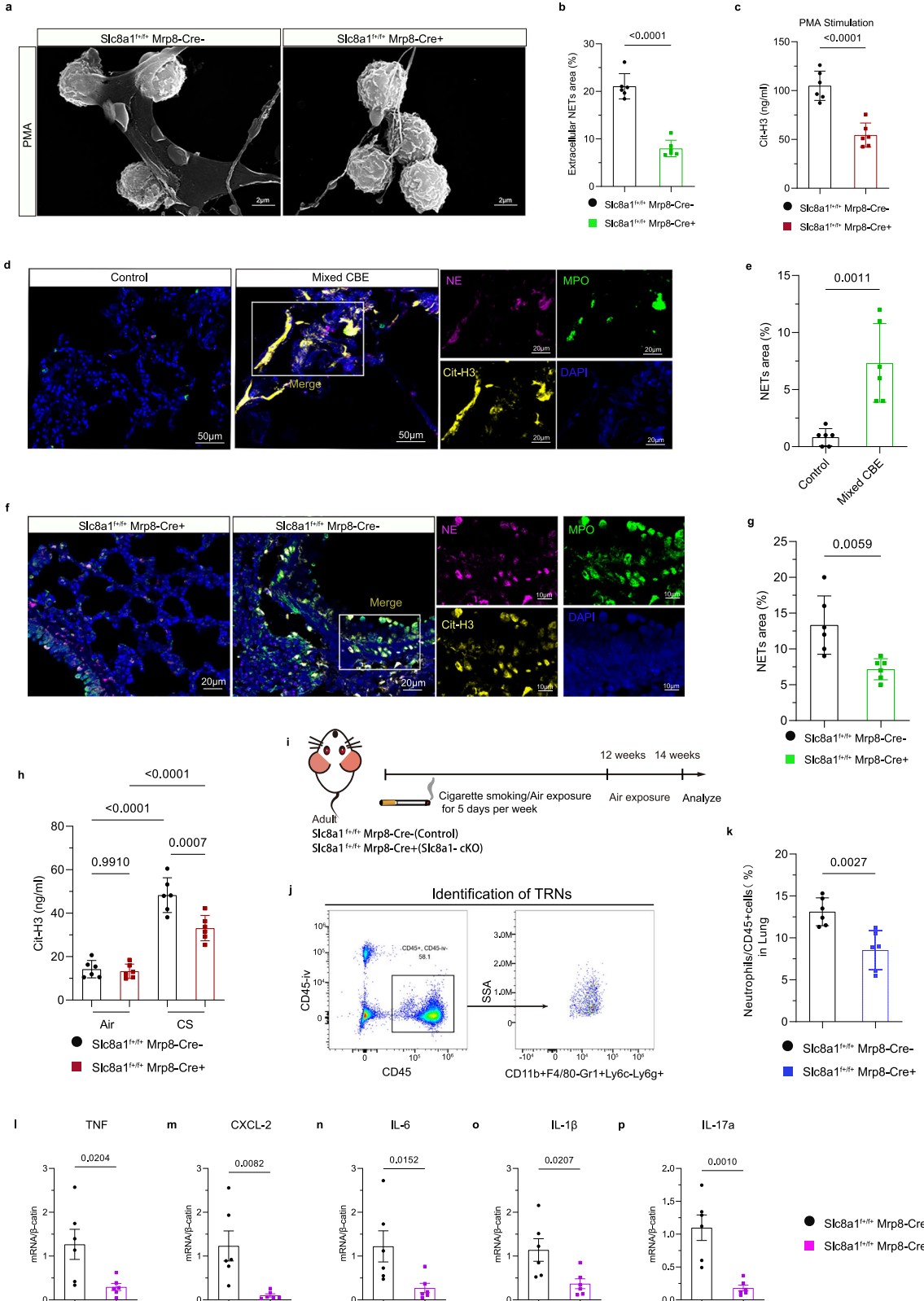

microenvironment that favors continued neutrophil recruitment and tissue retention, thus forming a self-perpetuating inflammatory loop to exacerbate COPD pathologies. NCX1 was positioned as an upstream regulator of this NET-driven loop. NCX1 inhibition might attenuates this loop, offering a potential therapeutic strategy to alleviate chronic neutrophilic inflammation in COPD. The present study provides a possible mechanistic insight into how NETs contribute to airway

remodeling and emphysema by mediating the aggregation and retention of neutrophils, highlighting the potential of targeting NCX1 as a therapeutic strategy to alleviate COPD-related lung damage. Compared to previous therapeutic attempts targeting CXCR2[34] or neutrophil elastase[47], which risk broader immunosuppression, targeting NCX1 offers a potentially more selective approach by specifically impairing the pathological NETosis process while preserving essential

**Fig. 5 | *Slc8a1* deficiency inhibited NETs formation of neutrophils. a** Purified bone marrow neutrophils are stimulated by PMA (50 nM) for 4 h at 37 °C to produce NETs, and are visualized by scanning electron microscope. Scale bar = 2 μm. **b** NETs area is calculated as the percentage of NETs to the total area (*n* = 6 mice). **c** ELIZA quantification of Cit-H3 levels in the cell culture supernatants of bone marrow-derived neutrophils with PMA stimulation (*n* = 6 mice). **d**, **e** Representative images of immunostained MPO (green), NE (purple), Cit-H3 (yellow) co-localization and quantification of NETs area in lung sections of control individuals and Mixed CBE patients (*n* = 6). Scale bar = 50 μm, 20 μm (magnified images). **f**, **g** Representative images of immunostained MPO (purple), NE (green), Cit-H3 (yellow) co-localization and quantification of NETs area in lung sections of CS-exposed *Slc8a1*[f+/f+];Mrp8-Cre[−] and *Slc8a1*[f+/f+];Mrp8-Cre[+] mice (*n* = 6 mice). Scale bar = 20 μm, 10 μm (magnified images). **h** ELIZA detection of Cit-H3 levels in BALF of different genotype mice under air or CS exposure (*n* = 6 mice). **i** Experimental scheme of *Slc8a1*[f+/f+];Mrp8-Cre[−] and *Slc8a1*[f+/f+];Mrp8-Cre[+] mice subjected to CS exposure for 12 weeks, followed by air exposure for 2 weeks. **j** FACS sorts and quantifies TRNs in the lungs of mice with different genotypes. **k** Quantification of TRN numbers in lungs of CS-induced mice with different genotypes after 2 weeks cessation of CS (*n* = 6 mice). **l–p** RT-qPCR detected mRNA levels of TNF, CXCL-2, IL-6, IL-1β, and IL-17a in *mouse* lungs with different genotypes (*n* = 6 mice) in Fig. 5i. Each data point represents one biologically independent replicate with three technical replicates (**b**, **c**, **e**, **g**, **h**, **l–p**). Data points in Fig. 5k represent biologically independent replicates. All quantitative data are presented as Mean ± SD. Two-sided *t*-test (**b**, **c**, **e**, **g**, **k**, **l–p**) and one-way ANOVA with Tukey's multiple comparison test (**h**) in this figure were used to calculate the *p*-values. At least 3 times, each experiment was independently repeated with similar results. Source data are provided as a Source Data file. CBE chronic bronchitis and emphysema; COPD chronic obstructive pulmonary disease; PMA Phorbol 12-myristate 13-acetate; NET neutrophil extracellular trap; Cit-H3 citrullinated Histone H3; CS cigarette smoke; TRNs tissue-resident neutrophils; ns no significance.

neutrophil functions. In summary, we identify NCX1 as a critical modulator of NET-mediated neutrophil retention and airway remodeling in COPD, offering a potential mechanistic framework and therapeutic target for chronic neutrophilic inflammation.

Calcium homeostasis is a fundamental regulator of immune cell function, particularly in neutrophils where tightly controlled $[Ca^{2+}]i$ dynamics govern activation, degranulation, and NETs formation[48]. In this context, NCX1 serves as a pivotal mediator of $Ca^{2+}$ flux across the plasma membrane[49]. NCX1 operates in two transport modes: the forward mode extrudes intracellular $Ca^{2+}$ in exchange for $Na^+$ influx, while the reverse mode imports extracellular $Ca^{2+}$ into the cytosol in exchange for $Na^+$ efflux[50]. While the forward mode is typically dominant under resting conditions, while the reverse mode activity triggered by pathophysiological stimuli results in extracellular $Ca^{2+}$ influx[50]. As a known trigger for inflammatory signaling, the pronounced intracellular $Ca^{2+}$ elevation enhances MAPK/NF-κB signaling, subsequently amplifying the production of inflammatory cytokines[51]. Our findings established that CSE potently activates NCX1 in its reverse mode, driving aberrant $Ca^{2+}$ influx into neutrophils. This influx is critical for initiating NETosis, a form of cell death characterized by the release of decondensed chromatin and granular proteins into the extracellular space. Previous studies have linked calcium influx with NETs formation, since the activation of calcium-dependent enzyme PAD4 is essential for citrullination of nuclear histones and chromatin de-condensation[52,53]. Earlier findings demonstrated that NCX1 in neutrophils can be modulated by intracellular $Na^+$ and $Ca^{2+}$ interplay, with increasing $[Na^+]i$ favoring reverse-mode NCX1 activation[23]. We further delineated that CSE and PMA might activate NCX1 reverse-mode not merely through $Na^+$ fluctuations but via a distinct and direct mechanism, characterized by simultaneous $[Na^+]i$ reduction and $[Ca^{2+}]i$ increase—suggesting potential membrane depolarization or alternative signaling events may be involved. Besides, CHO-K1 cells overexpressing recombinant NCX1 strengthened our conclusion that CSE induced an increase in $[Ca^{2+}]i$ through activation of NCX1 reverse mode. Our data demonstrated that $[Ca^{2+}]i$ elevation calcium overload as a key driver of NETs formation, as chelation of cytosolic $Ca^{2+}$ with BAPTA-AM suppressed, while $Ca^{2+}$ ionophore A-23187 strongly promoted Cit-H3 expression and NETs release. Of note, additional critical outcomes evidenced that pharmacological inhibition of NCX1 reverse-mode activity with KB-R7943 effectively suppresses NETs formation, preserves lung architecture, and improves pulmonary function in CS-induced COPD mice. The specificity of this intervention was confirmed by the reduced NETs release in NCX1 reverse-mode-inhibited neutrophils. Our study not only confirms this calcium dependency but also identifies NCX1 reverse-mode activity as the key route for CSE-induced $Ca^{2+}$ entry and downstream NETs release. These findings demonstrate that pathological NET formation in COPD is dependent on NCX1 reverse mode activation, and that calcium influx mediated by NCX1 constitutes a necessary and sufficient condition for driving this process. NCX1 reverse mode inhibition modulates NETs release indirectly via calcium entry, thereby preserving neutrophil viability and essential antimicrobial functions, which may reduce infection-related risks in clinical translation.

In conclusion, our study identifies NCX1 reverse-mode activity as a crucial mediator linking CS exposure to pathological $Ca^{2+}$ overload, NETs release, and subsequent tissue injury in COPD. Genetic deletion of *Slc8a1* or pharmacological inhibition of this transport mode not only halts NETs formation but also reverses hallmark features of COPD, including persistent airflow limitation, emphysema, airway remodeling, and alveolar stem cell depletion. These findings reveal a role for NCX1 in COPD and establish its reverse-mode activity as a promising, target-specific, and potentially safe therapeutic axis. This study has several limitations that should be acknowledged. First, NETs are reticular structures primarily produced by neutrophils to trap and kill pathogens; however, these structures may also bind to inflammatory cells other than neutrophils. Although our study demonstrated that inhibiting NETs mitigated COPD pathological changes, it is uncertain whether the observed airway remodeling and alveolar damage were directly attributed to NETs themselves or mediated by the retention and effects of other inflammatory cells. Second, it remains unclear which components of cigarettes directly interact with NCX1, causing a conformational change that triggers the reverse transport mode. This study did not explore the molecular mechanisms responsible for the transition between the reverse and forward modes of NCX1, and the precise mechanisms driving this mode switching remain a critical unresolved question in the field. Future in-depth work integrating high-resolution structural biology with real-time functional analyses and dedicated mechanistic approaches will be important to fully elucidate the molecular mechanisms governing NCX1's functional modes.

## Methods
### Human samples
The BALF and blood sample collection protocol was reviewed and approved by the Ethics Committee of the Affiliated Hospital of Zunyi Medical University (Approval No. KLL-2025-019). This study was conducted in strict accordance with the Declaration of Helsinki Ethical Principles for Medical Research Involving Human Subjects. The usage of clinical lung samples was approved by the Science and Technology Review Board and the Ethics Committee of Sichuan University (Sichuan, China, Ethics approval: IORG No. 20221582). Lung tissue samples were obtained from Clinical Biobank of Sichuan University West China Hospital (Sichuan, China). Lung specimens were collected from lung cancer patients, and only lung tissues from distal areas far from the cancer site were included in this study. The clinical characteristics and demographics of patients were shown in the Supplementary Table 1. Based on chest computed temography (CT) imaging,

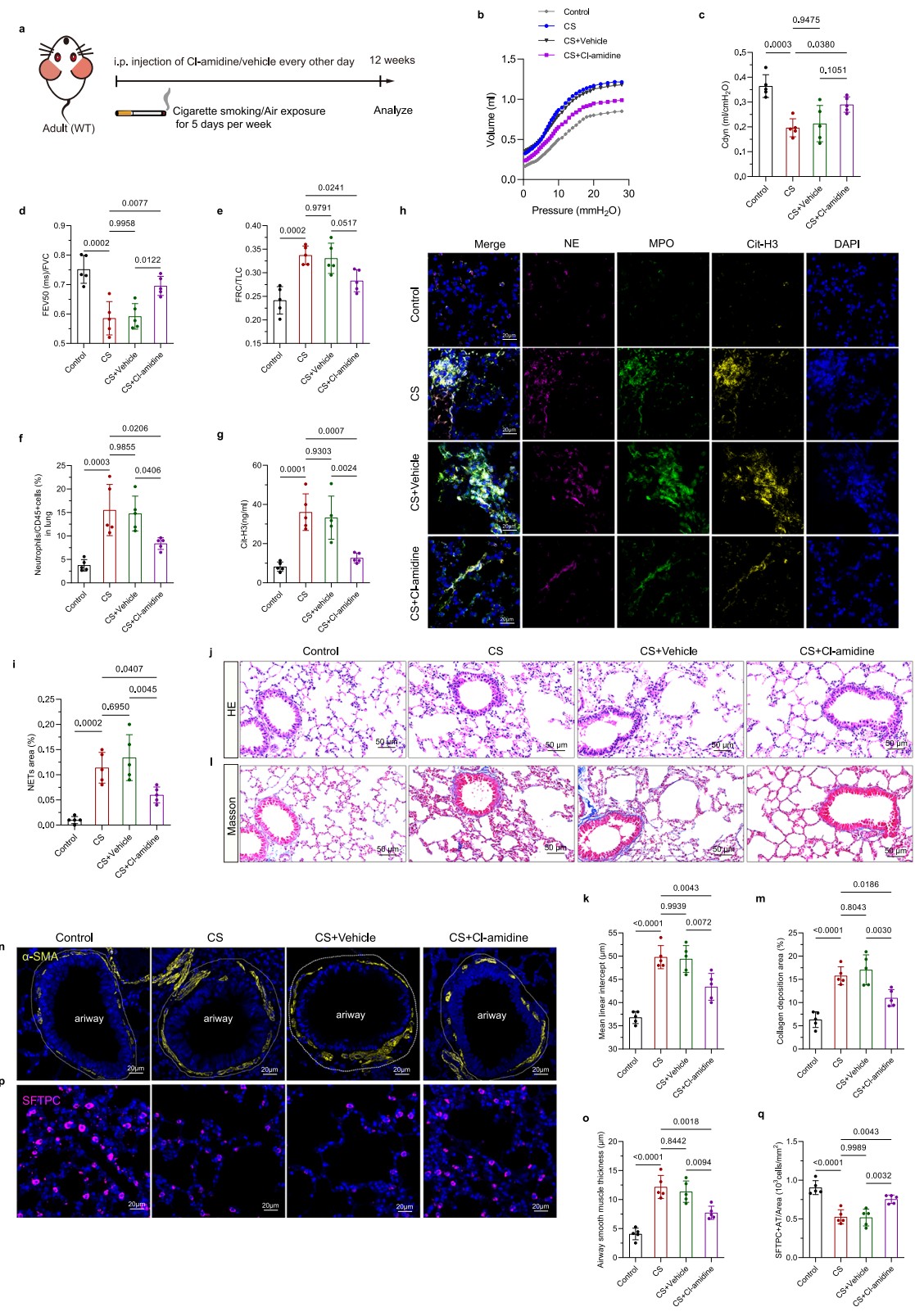

samples were stratified into 3 groups: Control group, normal lung tissue from individuals without chronic lung disease; Emphysema group, radiologically confirmed emphysema without chronic bronchitis features; Mixed CBE group, combined features of chronic bronchitis and emphysema. BALF samples were additionally collected from COPD patients and healthy controls. COPD patients were diagnosed based on the Global Initiative for Chronic Obstructive Lung

Disease (GOLD) guidelines for the diagnosis, management, and prevention of COPD. Between April and May 2025, 5 Control individuals (patients with asymptomatic pulmonary nodules) and 5 Mixed CBE patients with clinically diagnosed COPD and CT imaging were recruited from the Affiliated Hospital of Zunyi Medical University (Guizhou, China). Among the five healthy controls, four were non-smokers, and one was a smoker, while all five Mixed CBE patients had a history of

**Fig. 6 | NETs blockade by Cl-amidine attenuated neutrophils expansion and ameliorated pulmonary pathological changes in CS-exposed mice.**
**a** Experimental protocol of Cl-amidine treatment in WT mice during air or CS exposure. **b** The pressure-volume curves were generated in the Control, CS, CS +Vehicle and CS+Cl-amidine mice, presenting the mean values of volume measured under different airway pressures. **c**–**e** Pulmonary function measurements show dynamic compliance data, $FEV_{50(ms)}$/FVC and FRC/TLC ratios of Control, CS, CS +Vehicle and CS+Cl-amidine mice ($n = 5$ mice). **f** ELIZA detection of Cit-H3 levels in *mouse* BALF ($n = 5$ mice). **g** Flow cytometry quantification of neutrophil numbers in *mouse* lung tissues according to the experimental protocol in Fig. 2f ($n = 5$ mice). **h, i** Representative images of immunostained MPO (green), NE (purple), Cit-H3 (yellow) co-localization and quantification of NETs area in lung sections ($n = 5$ mice). Blue, DAPI-stained nuclei. Scale bar = 20 μm. **j, k** Representative H&E-stained lung sections showing alveolar destruction, and corresponding mean linear intercept values ($n = 5$ mice). Scale bar = 50 μm. **l, m** Masson's Trichrome staining and

quantification of lung collagen deposition area ($n = 5$ mice). Scale bar = 50 μm. **n**–**q** Immunofluorescence and quantification for α-SMA (yellow) and SFTPC (purple) showing smooth muscle hyperplasia and loss of alveolar stem cells ($n = 5$ mice). Scale bar = 20 μm. Each data point represents one biologically independent replicate with three technical replicates (**b**–**f, i, k, m, o, q**), and one biologically independent replicate (**g**). All quantitative data are presented as Mean ± SD. One-way ANOVA with Tukey's multiple comparison test was used to calculate the *p*-values for all quantitative data. At least 3 times, each experiment was independently repeated with similar results. Source data are provided as a Source Data file. CS cigarette smoke; Cdyn dynamic compliance; NE neutrophil elastase; MPO myeloperoxidase; Cit-H3 Citrullinated Histone H3; $FEV_{20(ms)}$ forced expiratory volume in 20 ms; $FEV_{50(ms)}$ forced expiratory volume in 50 ms; FVC forced expiratory volume; FRC functional residual capacity; TLC total lung capacity; ns no significance; HE Hematoxylin-Eosin staining; ns no significance.

smoking. BALF procedures were performed by experienced clinicians, and was processed immediately after collection for downstream analysis. A total of 50 mL of BALF was collected from each participant. In addition, 5 mL of peripheral venous blood was drawn from each individual for further analysis. The involved patients' information was provided in the Supplementary Table 2. All participants provided written informed consent prior to sample collection.

## Preparation of CSE
CSE was prepared using a standard protocol. Briefly, mainstream smoke from one cigarette was drawn using a peristaltic pump and bubbled through 10 mL of RPMI-1640 (Gibco, 11875101) at a constant flow rate. The resulting solution was adjusted to a neutral pH of 7.4, then filtered through a 0.22-μm filter to remove particulate matter and bacteria. The resulting solution was designated as 100% CSE stock. For experimental use, this stock solution was diluted to the indicated working concentrations (typically 10%) using fresh RPMI-1640 (Gibco, 11875101).

## FACS analysis of NCX1 expression on neutrophils in *human* BALF
BALF cells were filtered through 100 μm and 70 μm cell strainers to obtain a single-cell suspension. Red blood cell lysis was performed by incubating the suspension with RBC lysis buffer for 5 min at room temperature. The resulting cell pellet was resuspended in 200 μL of FACS buffer. For the detection of NCX1 expression, cells were incubated with anti-NCX1 antibody (Abcam, ab2869) at room temperature in the dark for 30 min. After incubation, the cells were washed twice with FACS buffer. Subsequently, the cells were stained with a combination of the following antibodies: PerCP/Cyanine5.5 anti-*human* CD45 (Biolegend, 368503), PE anti-*human* CD66b (Biolegend, 392903), APC/Fire750 anti-*human* CD11b (Biolegend, 301351), Brilliant Violet605 anti-*mouse* IgD (Biolegend, 405727). After incubation, the cells were washed twice with FACS buffer and analyzed by flow cytometer (BD FACSCanto II), and data were processed using FlowJo (v.10.0) software.

## Animal care
All mice used in the studies were housed under specific pathogen-free conditions at the Animal Center of Zunyi Medical University, with controlled temperature, humidity, and a 12-h light/dark cycle, and were provided with food and water ad libitum. *Slc8a1*flox/flox (*Slc8a1*f+/F+) mice and Mrp8 (S100a8) Cre recombinase (Mrp8-Cre) mice, both on a C57BL/6 background, were purchased from Cyagen Biosciences (Beijing, China). To generate neutrophil-specific *Slc8a1* knockout mice (*Slc8a1*f+/F+;Mrp8-Cre+), which carry a genomic DNA fragment deletion of exon 2 of *Slc8a1*, *Slc8a1*f+/F+ mice were crossed with Mrp8-Cre mice. The generated conditional knockout mice carried a loxP-flanked exon 2 of the *Slc8a1* gene, which is selectively excised in neutrophils via Cre recombinase expression under the Mrp8 promoter. Genomic DNA was extracted from the tail biopsies of transgenic mice and amplified with

specific primers to identify the *mouse* genotype. Primers used for genotyping were listed in Supplementary Table 3.

To establish COPD *mouse* models, 8–12-week-old male mice were passively exposed to CS generated from commercially available cigarettes (12 cigarettes/day, 4 times/day). Smoke exposure was carried out 5 days per week for a consecutive 12 weeks. All animal experiments were conducted in strict accordance with the Guidelines for the Care and Use of Laboratory Animals and approved by the Animal Ethics Committee of Zunyi Medical University (Approval No. ZMU20202409).

## Drug administration
WT mice received intravenous (i.v.) injection of KB-R7943 (1.5 μL of 100 mg/mL KB-R7943 diluted in 98.5 μL 0.9% saline per *mouse*, MCE, HY-15415) or Vehicle (1.5 μL DMSO in 98.5 μL 0.9% saline per *mouse*) every 2 days for total 12 weeks. In the experiments of Cl-amidine injection, WT mice were intraperitoneally (i.p.) injected of Cl-amidine (30 μL of 50 mg/kg Cl-amidine in 600 μL 0.9% saline per *mouse*, Selleck, 1043444-18-3) or Vehicle (30 μL DMSO in 600 μL 0.9% saline per *mouse*) every 2 days for total 12 weeks.

## Pulmonary function test
Pulmonary function was measured using a computer-controlled small animal ventilator system (DSI BuxcoPFT, America). Mice were deeply anesthetized with inhalation of isoflurane and placed in the supine position on a foam board. After tracheostomy and insertion of a cannula, mice were connected to the ventilator and respiratory functions were assessed. Standardized deep inflation maneuvers were performed to normalize lung volume history prior to measurements. Lung function parameters, including forced expiratory volume at 20 ms and 50 ms ($FEV_{20(ms)}$), $FEV_{50(ms)}$), forced vital capacity (FVC), functional residual capacity (FRC), and total lung capacity (TLC = inspiratory capacity + FRC), were measured. Data were analyzed using the manufacturer's software, and $FEV_{20(ms)}$/FVC, $FEV_{50(ms)}$/FVC, and FRC/TLC ratios were calculated to assess airway obstruction and hyperinflation. For pressure–volume loop analysis[54], the inflation and deflation maneuvers were performed by stepwise increasing airway pressure from 0 to 30 cmH$_2$O, followed by gradual release to baseline. The corresponding lung volume changes were continuously recorded to generate pressure–volume loops. From these curves, dynamic compliance (Cdyn) was calculated as the slope of the linear portion of the inflation curve using FinePointe software (DSI). The pressure–volume loops and Cdyn values were used to evaluate alterations in respiratory compliance under different experimental conditions.

## ELIZA detection of Cit-H3 in BALF and cultured neutrophils
For *mouse* BALF samples, 0.5 mL of cold PBS was instilled in fractions into the airways of mice. After allowing the PBS to stay in the lungs for 20 s, it was slowly withdrawn. This process was repeated three times to

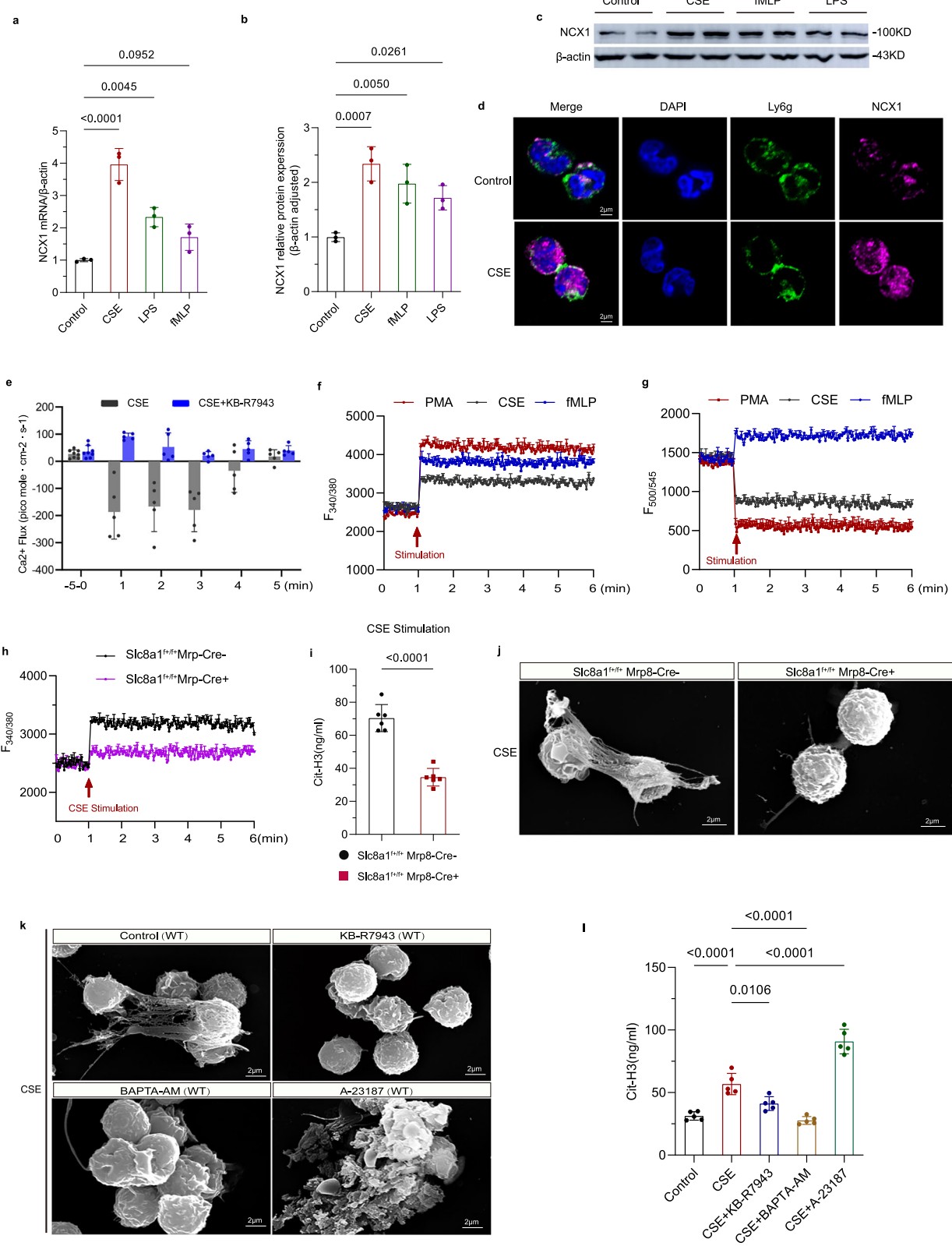

collect the lavage fluid. The collected BALF was centrifuged at 800–1000 × *g* for 10 min at 4 °C to separate the supernatants, which were stored at −80 °C for subsequent analysis. For neutrophils isolated from *mouse* bone marrow using flow cytometry, cells were first washed twice with BPS buffer, and $5 \times 10^5$ neutrophils were seeded into 24-well plates, followed by the indicated treatment (CSE, KB-R7943, BAPTA-AM, A-23187). The cells were incubated in a 37 °C incubator for 4 h.

After incubation, both cells and supernatants were collected. The supernatant was separated by centrifugation at 300 × *g* for 10 min at 4 °C. The levels of Cit-H3 in both BALF and cell culture supernatants were measured using the *Mouse* Guanylated Histone H3 ELIZA Kit (Jonlnbio, 48480), following the manufacturer's instructions. The absorbance was measured at 450 nm using a microplate reader, and

**Fig. 7 | NCX1 reverse-mode transport mediates Ca$^{2+}$ overload and NETs release in response to CSE. a–c** RT-qPCR and WB assess NCX1 expression in neutrophils upon CSE, fMLP and LPS stimulation ($n = 3$). **d** Immunofluorescence confirms NCX1 elevation indicated by Ly6g (green) and NCX1 (purple) in CSE-treated neutrophils ($n = 3$). Blue, DAPI-stained nuclei. Scale bar = 2 μm. **e** Real-time Ca$^{2+}$ flux dynamics in neutrophils with or without KB-R7943 administration measured by non-invasive micro-test technique ($n = 6$). Upward bars represent Ca$^{2+}$ efflux activity, and downward bars represent Ca$^{2+}$ influx. **f, g** Simultaneous real-time measurements of intracellular Ca$^{2+}$ (F340/380) and Na$^+$ (F500/545) in neutrophils by fluorescent probes upon stimulation with CSE, PMA, or fMLP ($n = 3$). **h** Ca$^{2+}$ influx in bone marrow-derived neutrophils from *Slc8a1$^{f+/f+}$*;Mrp8-Cre$^-$ and *Slc8a1$^{f+/f+}$*;Mrp8-Cre$^+$ mice upon CSE stimulation ($n = 3$). **i** ELIZA detects Cit-H3 expression in the cell culture of CSE-stimulated bone marrow-derived neutrophils ($n = 6$). **j** NETs formation in CSE-stimulated neutrophils is visualized by scanning electron microscope ($n = 3$). Scale bar = 2 μm. **k** Effects of KB-R7943, BAPTA-AM, and A-23187 on NETs area in CSE-stimulated neutrophils. Scale bar = 2 μm. **l** ELIZA detects Cit-H3 expression in unstimulated neutrophils and CSE-stimulated neutrophils with KB-R7943, BAPTA-AM, and A-23187 treatment ($n = 5$). Each data point represents one biologically independent replicate with 3 technical replicates (**a, b, l**). Data shown in Fig. 7e represent the mean of measurements (6 biological replicates). Data in Fig. 7f–h represent the mean of measurements (3 biological replicates with 3 technical replicates). All quantitative data are presented as Mean ± SD. Two-sided *t*-test (**i**) and one-way ANOVA with Tukey's multiple comparison test (**a, b, l**) were used to calculate the *p*-values. At least 3 times, each experiment was independently repeated with similar results. Source data are provided as a Source Data file. CSE cigarette smoke extract; PMA Phorbol 12-myristate 13-acetate; fMLP N-Formylmethionyl-leucyl-phenylalanine; LPS lipopolysaccharide; Cit-H3 Citrullinated Histone H3; WT wild type; min minutes; ns no significance.

the concentration of Cit-H3 in the samples was determined from a standard curve.

### Analysis of mean linear intercepts (MLI)

The right lung of mice was perfused with 1 mL of phosphate-buffered saline (PBS), followed by inflation with 4% paraformaldehyde (PFA) overnight at 4 °C for paraffin embedding. After fixation, the lungs were washed 3 times using cold PBS in 2 h at 4 °C, and were then dehydrated through a graded ethanol series (30%, 50%, 75%, 95%, and 100%). The dehydrated lungs were cleared with xylene for 1 h at room temperature, followed by incubation in paraffin at 65 °C for two 100-min cycles before being embedded in paraffin blocks.

To analyze the alveolar morphology, H&E staining was performed on paraffin-embedded lung sections. For each *mouse*, at least three randomly chosen sections were selected for examination. The MLI, an indicator of alveolar airspace enlargement, was calculated by dividing the total length of lines drawn across the lung section by the number of intercepts between the lines and alveolar walls, and analyzed using Image-Pro Plus software. The pulmonary vessels and large airways were excluded from the analysis.

### Masson's Trichrome staining

After dewaxing, the lung sections were stained with iron hematoxylin for 5 min, followed by differentiation with an acid ethanol solution to achieve a blue color. Then, they were washed with distilled water, and treated with Masson compound staining solution for 1 min, Ponceau magenta staining solution for 5 min, phosphomolybdate for 1 min, weak acid working solution for 5 min and aniline blue solution for 2 min. After each staining step, the sections were dehydrated in 95% ethanol, washed 3 times with xylene, and sealed completely with a neutral glue. Fibrotic area was visualized with a pathology Slide Viewing Software (Leica Biosystem) and analyzed using ImageJ_Fiji software.

### Multiplex immunofluorescence staining

Lung specimens and neutrophils were immunostained with a multiplex immunofluorescence kit (Afantibody, AFIHC037). The harvested *mouse* lungs were fixed in 4% PFA (Servicebio) for 24 h, washed with PBS, embedded in paraffin, and sectioned at 4-μm thickness. For deparaffinization, slides were immersed in xylene (3 × 10 min), dehydrated with graded ethanol (100% for 5 min; 95% for 5 min; and 70% for 5 min), and rinsed in distilled water for 2 min. Antigen retrieval was performed using antigen repair solution (pH 8.0, Servicebio) in the microwave at 75% power for 8 min, followed by 25% power for 7 min. Non-specific binding was blocked with 5% BSA for 35 min at room temperature. *Human* lung specimens were processed using the same protocol. Neutrophils sorted from *mouse* bone marrow or *human* BALF were seeded on coverslips coated with poly-L-lysine (Sigma, P4707), fixed with 4% PFA, washed with PBS, permeabilized with 0.3% Triton

X-100 in 5% BSA for 30 min at room temperature. The samples were then incubated overnight at 4 °C with primary antibodies: CD11c (Servicebio, GB11059, 1:200), CD68 (Proteintech, 25747-1, 1:200), CD3 (Servicebio, GB11014, 1:100), Ly-6G (Invitrogen, 14-5931-82, 1:50), αSMA (Invitrogen, 50976082, 1:200), CD66b (Novus Biologicals, NB100-77808, 1:100), SFTPC (Proteintech, 10774-1-AP, 1:200), MPO (Proteintech, 22225-1-AP, 1:200), NE (Abclonal, A8953, 1:200), Cit-H3 (Abways, CY6587, 1:250) and NCX1 (Abclonal, A5583, 1:200). Detailed information of primary antibodies was provided in Supplementary Table 4. Subsequently, slides were washed with PBS (3 × 5 min) and incubated with secondary antibodies (Poly horseradish peroxidase (HRP), *goat* anti-*mouse*/anti-*rabbit*, 1:200) for 50 min at room temperature. Fluorescent dye reaction solution was added to amplify the signals, and was washed 3 times with PBS. To enable multiplex staining, antibody stripping was performed by boiling tissue sections in antigen retrieval buffer (for tissue) or by incubating cell slides with antibody elution buffer (Afantibody, AFIHC038). Blocking and antibody incubation steps were repeated sequentially for each additional marker. Finally, nuclei were counterstained with DAPI (Beyotime, C1002). Three sections for each *mouse* were evaluated. Images were captured using a Zeiss LSM 980 microscope and analyzed using ImageJ_Fiji software. The quantification of the NET area was determined based on the colocalized regions of MPO and Cit-H3 signals.

### Immunohistochemistry (IHC)

Immunohistochemistry was performed to detect the expression of MPO and NE in *mouse* lung tissues. The prepared sections were deparaffinized in xylene (3 × 10 min), rehydrated through graded ethanol (100%, 95%, and 70%, each for 5 min), and rinsed in distilled water. Antigen retrieval, blockade of endogenous peroxidase activity and non-specific binding were followed. Sections were then incubated overnight at 4 °C with the following primary antibodies: MPO (Proteintech, 22225-1-AP, 1:200) and NE (Abclonal, A8953, 1:200). Detailed information on primary antibodies was provided in Supplementary Table 4. After PBS washing, HRP-conjugated secondary antibody (Servicebio, 1:200, *goat* anti-*rabbit*) was applied for 50 min at room temperature. Signal was developed using DAB substrate (Servicebio, G1212), and nuclei were counterstained with hematoxylin. Slides were dehydrated, cleared, and mounted using neutral resin. Three sections per *mouse* were evaluated. Images were captured using a Zeiss LSM 980 microscope and analyzed using ImageJ_Fiji software.

### Neutrophil isolation for in vitro experiments

Bone marrow neutrophils were isolated from C57BL/6 *mice* aged 12 weeks with genotypes confirmed by PCR. Cells were washed with sterile Hanke's Balanced Salt Solution (HBSS, Invitrogen, 14185052), and incubated with CD45-APC-Cy7 (BD, 557659), CD11b-BB515 (BD, 564454), and Ly6g-PE (BD, 551461) antibodies. Neutrophils (CD45$^+$CD11b$^+$Ly6g$^+$ cells) were then isolated by flow cytometry.

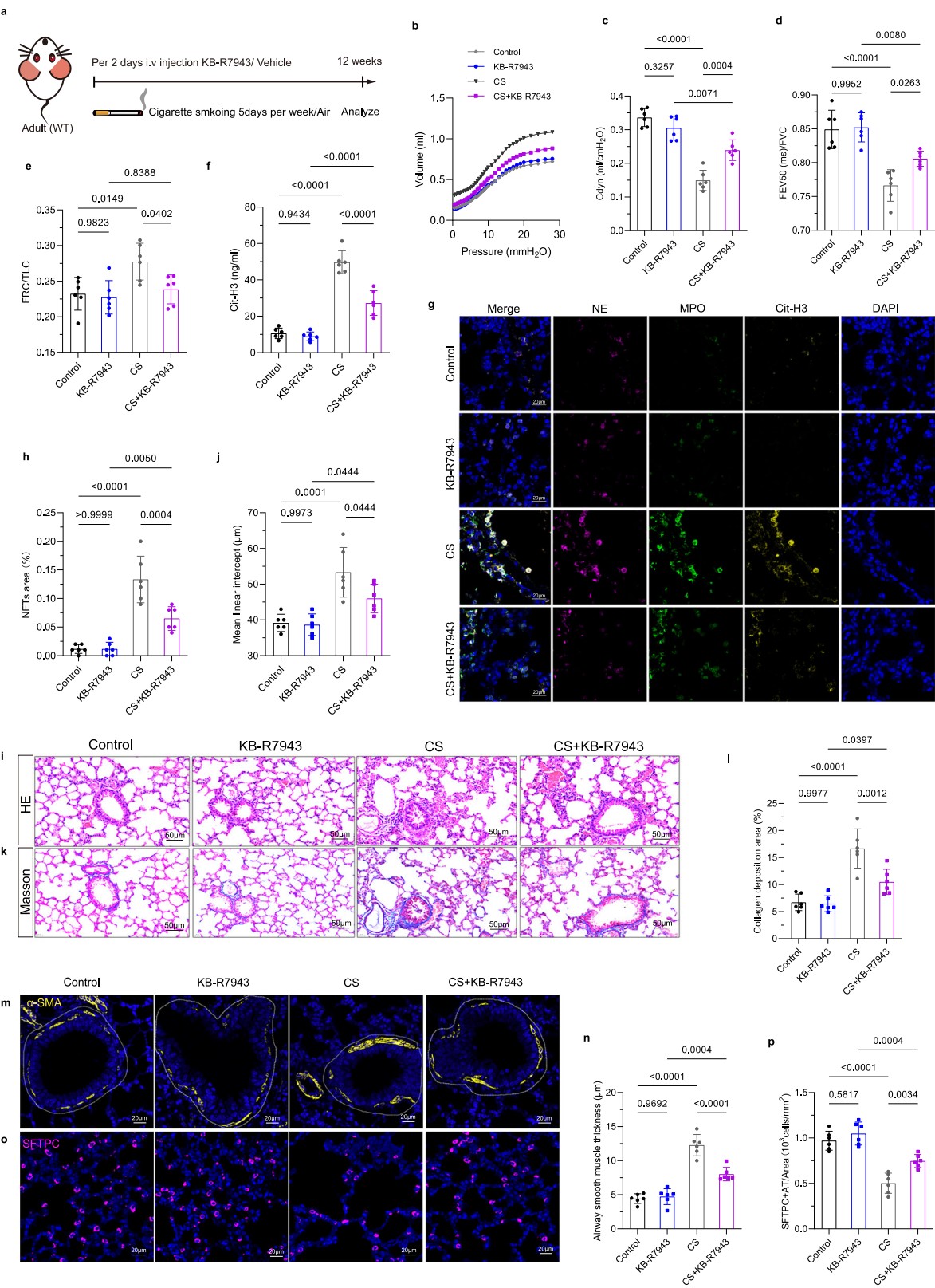

Detailed information on antibodies was provided in Supplementary Table 4. The isolated neutrophils were verified by flow cytometry, with 95−98% purity. *Human* neutrophils were isolated from the peripheral blood of COPD patients and healthy controls using the *Human* Peripheral Blood Neutrophil Isolation Solution kit (Solarbio, P9040), following the manufacturer's protocol. Neutrophils in *human* BALF were sorted as CD45$^+$CD11b$^+$CD66b$^+$ cells by FACS.

**Tissue dissociation and flow cytometry**

A digestion cocktail including DNase (50 U/ml, Sigma), Dispase (15 U/ml, Beyotime), and Collagenase Type I (225 U/ml, Servicebio) was tracheally perfused into the *mouse* lung, followed by perfusion with warm PBS through the right ventricle. The lungs were then subsequently removed from the chest. To prepare single-cell suspensions for FACS analysis, the lungs were digested using a cocktail consisting of DNase (40 U/ml,

**Fig. 8 | Pharmacological inhibition of NCX1 reverse mode by KB-R7943 protects against CS-induced COPD. a** Schematic overview of KB-R7943 administration in mice during CS or air exposure. **b** The pressure-volume curves were generated in the mice of Control, KB-R7943, CS, and CS + KB-R7943 groups, presenting the mean values of volume measured under different airway pressures. **c–e** Pulmonary function tests measure dynamic compliance data, $FEV_{50(ms)}$/FVC and FRC/TLC ratios ($n = 6$ mice). **f** ELIZA detection of Cit-H3 levels in *mouse* BALF ($n = 6$ mice). **g, h** Representative images of immunostained MPO (green), NE (purple), Cit-H3 (yellow) co-localization and quantification of NETs area in lung sections ($n = 6$ mice). Blue, DAPI-stained nuclei. Scale bar = 20 μm. **i, j** Representative H&E-stained lung sections illustrate alveolar structure, and corresponding mean linear intercept values ($n = 6$ mice). Scale bar = 50 μm. **k, l** Masson's Trichrome staining and quantification of lung collagen deposition area ($n = 6$ mice). Scale bar = 50 μm. **m–p** Immunofluorescence and quantification for α-SMA (yellow) and SFTPC (purple) shows smooth muscle hyperplasia, and loss of alveolar stem cells ($n = 6$ mice). Scale bar = 20 μm. Each data point represents one biologically independent replicate with three technical replicates (**b–f, h, j, l, n, p**). All quantitative data are presented as Mean ± SD. One-way ANOVA with Tukey's multiple comparison test was used to calculate the *p*-values. At least 3 times, each experiment was independently repeated with similar results. Source data are provided as a Source Data file. CS cigarette smoke; Cdyn dynamic compliance; NE neutrophil elastase; MPO myeloperoxidase; Cit-H3 Citrullinated Histone H3; $FEV_{50(ms)}$ forced expiratory volume in 50 ms; FVC forced expiratory volume; FRC functional residual capacity; TLC total lung capacity; HE Hematoxylin-Eosin staining; ns no significance.

Sigma) and Liberase TM (80 mg/mL, Roche), diced with tissue scissors, and further processed using the gentle MACS Octo Dissociator with Heaters (Miltenyi Biotec, 130096427). The obtained suspensions were washed with FACS buffer and passed through 70 μm and 40 μm cell filters, respectively. After centrifugation, the precipitation was re-suspended with RBC lysis buffer (Thermo Fisher, NC9067514), incubated for 3 min, centrifuged again, and finally washed twice with FACS buffer to remove residual lysis buffer. For surface marker staining, cell suspensions were incubated with appropriate antibodies in FACS buffer for 45 min at 4 °C, followed by washes with FACS buffer. The antibodies used for staining included: CD45-APC-Cy7 (BD, 557659), CD11b-BB515 (BD, 564454), Ly6g and Ly6C-APC (BD, 553129), Ly6g-PE (BD, 551461), Ly6C-BV605 (BD, 563011), F4/80-BV421 (BD, 565411). For FACS analysis of neutrophil development, antibodies Ly6g/Ly6C-FITC (Proteintech, 65140), CD11b-APC (Biolegend, 101211), and CD117(c-kit)-PE (Biolegend, 161503) were used. Detailed information on antibodies was provided in Supplementary Table 4. For FACS analysis of neutrophil phagocytosis, neutrophils were incubated with pHrodo Red-conjugated Escherichia coli (Invitrogen, P35361). For distinguishing the circulating and resident neutrophils, 3 mg of Alexa Fluor 700-conjugated CD45 antibody (Biolegend, 147715) diluted in 100 μL PBS (1×) was intravenously injected via the tail vein 3 min before euthanasia. The stained samples were analyzed using a flow cytometer (BD FACSCanto II), and data were processed using FlowJo (v.10.0) software.

### Real time quantitative PCR (RT-qPCR)
Total RNA was extracted from whole lungs or neutrophils isolated by flow cytometry using PicoPure RNA Isolation Kit (Applied Biosystems, KIT0204), following the protocols provided by the manufacturer. The SuperScript RT Kit (Invitrogen, 18080044) was used to reverse-transcribe total RNA into cDNA. PCR amplification was performed using the qPCR SYBR Green Master Mix (Yeasen, 11201ES08). Gene-specific primers for NCX1, MPO, NE, TNF, CXCL-2, IL-6, IL-1β, and IL-17a were designed by Youkang Biotech and listed in Supplementary Table 3. GAPDH was used as the internal control. Relative expression levels were calculated using the $2^{-\Delta\Delta Ct}$ method with normalization to GAPDH.

### Western blot
Lung tissues and isolated neutrophils were homogenized in RIPA buffer containing a mixture of protease and phosphatase inhibitors Cocktail (Sigma-Aldrich, PPC1010). Total protein concentration was determined using BCA Protein Assay Kits (Thermofisher, 23227). Equal amounts of protein from each sample were loaded onto a PAGE Gel Quick Preparation Kit (Yeasen, 20325ES05), and subsequently transferred to PVDF membranes (Yeasen, 36125ES03, 36126ES03). After being blocked with QuickBlock Blocking Buffer (Beyotime, P0252FT), the membranes were incubated with primary and secondary antibodies successively. The following antibodies were used: anti–NE (Abclonal, A8953, 1:1000, rabbit), anti–MPO (Proteintech, 22225-1-AP, 1:5000, rabbit), anti–Cit-H3 (Abways, CY6587, 1:5000, rabbit), anti-

*Slc8a1* (Abclonal, A5583, 1:5000, rabbit), anti–β-actin (CST, 4967S, 1:1000, rabbit), and secondary antibodies (Goat Anti-Rabbit IgG, Abcam, ab6702, 1:5000). Detailed information of primary antibodies was provided in Supplementary Table 4. Protein bands were visualized using Amersham Imager 600 system, and blots images were imported into the ImageJ_Fiji software for densitometric analysis and quantification.

### Measurement of real-time $Ca^{2+}$ flux using non-invasive micro-test technology (NMT)
Neutrophils were isolated from bone marrow and resuspended in testing solution (0.4 mM $Ca(NO_3)_2$, 100 mM NaCl, 5.0 mM $Na_2HPO_4$, 5.0 mM KCl, 1.0 mM HEPES (pH 7.0)). The cells were incubated in this solution for 10 min before measurement. To measure $Ca^{2+}$ flux, neutrophils were placed under a microscope and targeted cells were identified. A $Ca^{2+}$ flux microsensor (NMT Physiolyzer, YoungerUSA) was positioned 5 μm from the target cell to begin the measurement. Baseline $Ca^{2+}$ flux data were recorded for 5 min for each cell. Subsequently, CSE or CSE + KB-R7943 stock solutions were added to the testing environment until working concentrations were achieved. Immediate post-treatment $Ca^{2+}$ flux data were then recorded for an additional 5 min. Each experimental group consisted of 6 single-cell measurements (6 biological replicates). $Ca^{2+}$ flux data were read and exported directly using imFluxes V3.0 software (Xuyue Company). The flux rate was expressed in units of pico mol • $cm^{-2}$ • $s^{-1}$. The sign of the flux value indicates the direction of $Ca^{2+}$ transport: a positive value represents the efflux of $Ca^{2+}$ from the cell, whereas a negative value represents the influx into the cell.

### CHO-K1 cell culture and transfection
CHO-K1 cells (IMMOCELL, China) were cultured in F12-K media (Gibco, USA) supplemented with 10% fetal bovine serum (FBS) at 37 °C in a humidified 5% $CO_2$ incubator. To overexpress NCX1, CHO-K1 cells were transfected with plasmids carrying the cDNA of *Slc8a1* (*mouse*), purchased from Miaoling Plasmid (https://en.miaolingbio.com/). The plasmids were transfected using Lipofectamine 2000 (Invitrogen, USA) according to the manufacturer's instructions. To select successfully transfected cells, G418 (Sigma-Aldrich, USA), an antibiotic for positive selection, was applied to the cells. The effective concentration of G418 was determined by a preliminary titration to inhibit 100% of the non-transfected cells. Transfected cells were cultured in the presence of G418 for approximately two weeks to allow the selection of stable clones. To confirm the successful transfection of NCX1 plasmids, WB and immunofluorescence were used to confirm NCX1 expression.

### Measurements of $[Ca^{2+}]i$ and $[Na^+]i$ using fluorescent ion indicators
*Mouse* neutrophils were sorted from bone marrow using flow cytometry ($CD45^+CD11b^+Ly6g^+$). Approximately $5 \times 10^5$ neutrophils were used per experiment. Prior to measurement, the cells were washed

twice with Ringer's solution (1.2 mM CaCl$_2$, 140 mM NaCl, 5.4 mM KCl, 0.8 mM MgCl$_2$, 5.5 mM glucose, 10 mM HEPES, pH 7.4). Neutrophils were then loaded with 3 µM Fura-2 AM (Maokangbio, MX4502) or 5 µM NaTrium Green-2 AM (Maokangbio, MX4514), both containing 0.02% Pluronic F-127 (Maokangbio, MX4502), and incubated for 30 min at 37 °C. After incubation, the cells were washed twice with Ringer's solution and further incubated at 37 °C for an additional 30 min to ensure complete de-esterification of the AM esters. Intracellular [Ca$^{2+}$]i and [Na$^+$]i levels were measured using a Multifunctional Microplate Detector (Agilent, USA). Basal [Ca$^{2+}$]i and [Na$^+$]i concentrations were recorded from 0 to 60 s. Subsequently, the following reagents were sequentially added using an automatic drug sampling system: CSE (10%), PMA (50 nM), fMLP (10 µM), CSE+Vehicle, CSE + KB-R7943 (10 µM), BAPTA-AM (10 µM), A-23187 (10 µM). The [Ca$^{2+}$]i and [Na$^+$]i levels were measured from 61 to 360 s. Fura-2 AM emission ratios were used to quantify the integrated [Ca$^{2+}$]i signal (F340/F380) and [Na$^+$]I (F500/545). To exclude the potential involvement of mechanosensitive ion channels, an additional set of experiments was performed using GsMTx4 (MCE, HY-P1410), a specific inhibitor of Piezo channels. Neutrophils were preincubated with 2.5 µM GsMTx4 for 10 min at 37 °C before stimulation with CSE, PMA, or fMLP, and subsequent [Ca$^{2+}$]i and [Na$^+$]i fluxes were recorded under the same experimental conditions as described above.

### NETs formation in vitro and scanning electron microscope observations

To visualize NETs formation, neutrophils ($5 \times 10^5$ cells) were resuspended in RPMI-1640 (Gibco, 11875101) and were seeded on cell culture slides (Biosharp, BS14RC) in 24-well plates. The cells were incubated at 37 °C for 4 h, and were then treated with PMA (50 nM, Selleck), CSE, and/or Cl-amidine (100 mM, Selleck) to induce NETs formation. Following the conventional dehydration and sputter coating procedures, the NETs were observed using a scanning electron microscopy (ZEISS, EVO10) at an accelerated voltage of 30 kV. Three sections per group were selected for the quantification of the NETs area.

### Statistical analysis

Statistical analysis was performed using GraphPad Prism software (v.9.0). Data were assessed for normality using the Shapiro-Wilk test. Continuous variables were described as median (interquartile range, IQR) or mean (standard deviation, SD). Categorical variables are described as proportions. Student's $t$-test (for normally distributed data) or Mann-Whitney U-test (for non-normally distributed data) was used for comparisons of continuous quantitative variables between two groups. Proportional differences were compared using the chi-square test (or Fisher's exact test). For comparisons among three or more groups, ANOVA was employed. Correlation of NCX1 expression with levels of NE and MPO was evaluated using Spearman's correlation analysis (nonparametric data). The significance of differences between the two groups was determined using a two-sided unpaired $t$-test. Among three or more groups, one-way ANOVA with Tukey's multiple comparison test was used for comparisons. A $p$-value less than 0.05 was considered statistically significant for all analysis.

### Reporting summary

Further information on research design is available in the Nature Portfolio Reporting Summary linked to this article.

## Data availability

All original data generated or analyzed in this study are provided in the article and its Supplementary files or from the corresponding author upon request. Source data are provided with this paper.

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

## Acknowledgments

We express thanks to Xiu-Ying He and Fei Liu from Sichuan University West China Hospital for their assistance with clinical sample obtainment. We would like to thank Meng-Li Zhu, Xiang-Yi Ren, Jian Yang, Dan Li, Hong-Ying Chen, Yan Wang, Hui-Fang Li, Cong Li, Bei-Bei Liu, Jing-Yao Zhang, and Li-Wen Qin (Cytology and Molecular Platform, Core Facilities of West China Hospital, Sichuan University) for their technical guidance. We thank Tao Wang and Jun Chen from the Laboratory of Pulmonary Diseases at West China Hospital of Sichuan University for conducting the pulmonary function tests on mice. We appreciate platform support from the Guizhou Provincial Key Laboratory of Pathogenesis and Prevention of Common Chronic Diseases. This study was financially supported by the National Natural Science Foundation of China (No. 82060016; SXL) and Postgraduate Scientific Research Funding Project of Guizhou Province (No. 2024YJSKYJJ340; JQX).

## Author contributions

T.H.W., Y.X., Y.O., and S.X.L. conceptualized and supervised this study. The experiments were performed by S.X.L., Y.W.W., L.M.S., H.Y.H., Y.T.L., and P.P.S. Data were processed, analyzed and visualized by S.X.L., L.Y.Z., J.Q.X., and J.Z. The manuscript was drafted by S.X.L. and X.A., and was reviewed and revised by S.X.L., L.C., Y.X., and Y.O. All authors discussed results and approved the final version of the manuscript.

## Competing interests

The authors declare no competing interests.
