## [Transparent Peer Review file · Nature Communications]

NCX1 reverse mode promotes calcium-dependent NETs formation and lung damage in chronic obstructive pulmonary disease

Corresponding Author: Dr Shi-Xia Liao

Version 0:

Reviewer comments:

Reviewer #1

(Remarks to the Author)

The manuscript by Liao and coll. postulates that the reverse mode of NCX1 (Slc8a1) sodium/calcium exchanger is responsible for neutrophil-mediated lung damage in mice under conditions mimicking chronic obstructive pulmonary disease (COPD). Authors found enhanced expression of NCX1 in human samples from COPD individuals as well as from mice exposed to cigarette smoke. Authors propose that NCX1 acting in reverse mode, thus mediating calcium influx, is responsible for enhanced release of neutrophil extracellular traps (NETs). To support their claim, authors generate a conditional mouse model with selective ablation of NCX1 expression in neutrophils. This mouse model shows reduced formation of NETs upon stimulus and decreased COPD signs. However, the study lacks several control experiments that are needed to support the conclusions, particularly regarding the direct involvement of NCX1-mediated calcium influx in promoting altered neutrophil function,

SPECIFIC POINTS

Figure 1 A. The specificity of the NCX1 antibody used for immunofluorescence in murine and human samples should be demonstrated by disappearance of the signal in neutrophils from KO animals.

Supplementary Figure 2E. The expression of NCX1 detected by immunoblot in neutrophils from KO animals is markedly decreased but not absent. Is this a sign of a residual expression/function of the transporter?

Figure 7. In my opinion, the claim that NCX1 is working in reversed mode and promoting high intracellular concentration is not adequately supported by experiments. In particular, the reverse mode of NCX1 is not an intrinsic characteristic of the transporter but depends on the ion gradients across the plasma membrane. An increase in intracellular sodium concentration can reverse the activity of NCX1 in neutrophils (Najder et al. Role of the intracellular sodium homeostasis in chemotaxis of activated murine neutrophils. *Front Immunol* 11:2124, 2020). Therefore, the effect of cigarette smoke may be indirect, increasing intracellular sodium. This issue has not been investigated. To study NCX1 function, authors use an unconventional method, based on a sensor that supposedly measures calcium flux direction. Looking at the vendor website, this instrument is mostly used for ion flux in plants. There are no tests in the paper to demonstrate that the method is really working compared to classical methods (e.g. measurement of intracellular calcium with ion-sensitive fluorescent probes). For example, there is no experiment showing that the calcium flux is abolished in neutrophils from NCX1 KO mice. Also, the NCX1 inhibitor should also be tested in neutrophils from KO animals to show that it does not alter flux when NCX1 expression/function is ablated. Additional experiments should have used methods to buffer or increase intracellular calcium by other means (e.g. calcium-chelating agents, calcium-mobilizing agonists, calcium ionophores) to check the effect on NET release and/detection of calcium flux by the sensor.

Reviewer #2

(Remarks to the Author)

Major strengths:

1. Fascinating findings supporting a novel mechanism in which the NCX1 transporter reverses its mode in neutrophils resulting in altered intracellular calcium and triggering NETosis.

2. The correlations in Figure 1I are very nice.
3. Beautiful SEM pictures throughout the manuscript.
4. The data shown in Figure 6 strengthened the manuscript.
5. Strengthened by both human and murine data.
6. Further made stronger by using multiple approaches to prove their mechanism (gene deletion, pharmacologic inhibition).

Major critiques:

1. The nomenclature used of "simple emphysema" vs "COPD" is confusing and not consistent with clinical nomenclature as emphysema is a form of COPD. Would suggest using the term "mixed chronic bronchitis emphysema" vs "emphysema" to differentiate the two, if that is in fact what they mean.
2. Figure 1C lists mRNA changes between "emphysema" and their "COPD" group, but the protein results for NCX1, MPO or NE in their "emphysema" group are not shown. Showing those results would strengthen their findings that the "simple emphysema" vs their "COPD/mixed chronic bronchitis emphysema" groups are molecularly distinct.
3. Figure 1A: where in the lung are these tissue samples from? Hard to discern underlying lung structure. Perhaps labels of alveoli, etc would improve this.
4. Figure 1K: would be greatly strengthened if they could isolate neutrophils vs other immune cell subsets from patient BAL and run western blots for NCX1 expression. This may also be demonstrated by flow cytometry if NCX1 can be stained for by flow. The microscopy alone on lung tissue is insufficient to substantiate their statement that NCX1 is much higher in neutrophils vs other immune cells in control vs COPD patients.
5. Are elevations in neutrophil NCX1 expression limited to lung neutrophils in COPD patients or is NCX1 expression increased in peripheral blood too?
6. Demonstrate that Slc8a1 deletion was neutrophil specific by isolating other immune cell types (mono/macros, etc) from these animals and showing NCX1 protein remains present in similar concentrations to control animals.
7. Figures 3/4: While demonstrating that neutrophil numbers in each compartment are similar in both genotypes and showing that neutrophil function of bone marrow neutrophils are similar in both genotypes are solid starts, particularly as it relates to #5 above, would suggest testing functions of lung neutrophils between both genotypes as well.
8. Figure 5: In addition to microscopy, would be much stronger if use an additional method to quantify NETs to substantiate the authors conclusions. For example, could measure NE: DNA, MPO:DNA or CitH3: DNA complexes by dual-ELISAs or Cit-H3 ELISA in the murine BAL. Same critique applies to Figure 7F & H.
9. Insufficient evidence to substantiate the proposed mechanism re NETs impeding the normal regression of tissue resident neutrophils from the lung, described in lines #191-193. There are several potential explanations including NETs promoting the release of chemokines. This has been previously demonstrated in lung epithelium by the authors of PMID # 36703990 and in fibroblasts by the authors of PMID # 23536012. Could measure the chemokine and cytokine levels in the BAL and/or whole lung to substantiate this as a mechanism.
10. Figure 6: Need to confirm the Cl-amidine treatment decreased NET generation as they expect by microscopy or ELISA, etc.
11. For Figure 7I: are there microscopy pictures that coincide with the graphs enumerating changes in the lungs?
12. The authors use PMA to induce NETs for comparison. While PMA is a robust stimulator of NETs, would suggest they also compare to more physiologically relevant stimuli such as LPS or IL-8 (Figure 2,5 and 7).
13. Likely, outside the scope of the current study, however, showing that the Slc8a1 deleted mice have improvement in their pulmonary function tests after CSE would be an outstanding demonstration of the functional impact of NCX1 transporter in a COPD model.
14. Discussion needs to be strengthened by a deeper discussion of how these findings are consistent or inconsistent with prior studies, providing enhanced context for the current findings and how the results truly advance the field.

Minor critiques:

1. Abstract: several typos and minor grammatical errors need correcting
2. Abstract: write out what MPO and NE are abbreviations for
3. This statement as it currently reads, is incorrect, lines #63-64: "...COPD patients may suffer from drug induced neutropenia..." That is not a known lab finding in COPD unless they are referring to result of a specific therapy that is not widely used? Also a reference is needed for this statement.
4. Figure 1A legend or the text: indicate what immune cell types that markers listed in green are meant to identify, e.g. CD3= T cells. Of note, both CD66b and CD11c are present on neutrophils.
5. In section 2.2, indicate that the Slc8a1 is the gene for NCX1.
6. Clarify if NET area is measured where there is overlap of at least two relevant markers (Figure 4).
7. Section 2.4: Note Cl-amidine is a pan-PAD inhibitor and is not specific for PAD4.

Reviewer #3

(Remarks to the Author)

In the manuscript of Liao et al., the investigators characterized the role of NCX1 in neutrophils and its contribution to the pathogenesis of COPD.

They utilized in vitro and in vivo models including human samples showing significant upregulation of NCX1 in neutrophils, which correlated with MPO and NE. Furthermore, they showed attenuation of the neutrophil response in COPD including decreased emphysema in smoke-exposed mice with a neutrophil specific NCX1 deletion. Furthermore, cigarette smoke led

to NCX1 to operate in reverse transport mode causing increased intracellular calcium and abnormal NET formation. Inhibition of these processes also attenuated CS-induced chronic bronchitis and development of emphysema like changes. Overall, this manuscript describes the involvement of neutrophil NCX1 in the COPD lung in addition to elucidation the downstream signaling pathways that are mediating the lung changes and neutrophil response thereby describing a potential novel pathway to target in the neutrophilic COPD phenotype. There are some moderate concerns that should be addressed.

- 1) Supplemental figure 1A will need quantification of airway disease (MLI etc).
- 2) Fig. 2D-J, control mice are missing, that are not exposed to cigarette smoke.
- 3) Fig. 3 says that there was no difference in neutrophil number, but Fig. 2C shows the difference in number in the lung – this should be explained/it is not made clear whether these mice were exposed to cigarette smoke in Fig 3 (if not, it would be hard to measure migration to the lung if not stimulated). This is definitely a group that should be included.
- 4) Fig. 6 G does not correlate with the images and should be shown with better representative images.
- 5) The PAD4 inhibitor should be better described and should be validated. Neutrophils are not the only cells that express PAD4 and this should be taken into consideration.
- 6) FlexiVent analysis of these mice would be helpful.
- 7) Please comment on how many mice (male, female) were used and how many total donors and replicates in the Methods Statistics section.

Minor:

- 1) The PAD4 inhibitor and specificity for NETs should be introduced and validated.
- 2) The neutrophil isolation should be described in more detail: in the methods section, it is said that they were isolated from BALB/c mice, but the models used are all C57BL/6
- 3) Were neutrophils isolated via flowcytometry sorting?

Version 1:

Reviewer comments:

Reviewer #2

(Remarks to the Author)

Authors provide a thoughtful and thorough response addressing all of my critiques and concerns.

One important clarification needs to be made: The authors refer to measuring IL-8 in mice in Figure 5, but mice do not express IL-8, which is a bit concerning. The murine homologues are CXCL-1, CXCL-2 and CXCL-5 and are important neutrophil chemoattractants. What exactly did the authors measure? Did they use murine primers or human primers for this RT-PCR?

I would strongly consider showing pressure-volume loops which they should be able to generate and would be visually more clear as the FEV1(20) and (50) are not used commonly used clinically. The authors likely also have compliance and elastance data too that would strengthen their findings as well of emphysema or lack thereof.

Reviewer #3

(Remarks to the Author)

All concerns have been appropriately addressed.

Reviewer #4

(Remarks to the Author)

Dear ladies and gentlemen,

I was asked to re-review new data and comments given to address the concerns of reviewer #1 following the initial submission of the manuscript.

Reviewer #1 only had three specific points, and the authors have included several new data to address those:

Point 1 - Specificity of the NCX1 antibody - The authors now very clearly demonstrate the specificity of the NCX1 antibody, the new data are very convincing.

Point 2 - Remaining expression of NCX1 in neutrophils from knockout animals - The authors now demonstrate that the purity of neutrophils following FACS is "only" 95%-98%. Thus, 2-5% of the cells are other immune cells that still express NCX1. This is convincing and sufficient to address this concern.

Point 3 - This point is more "major" than point 1 and 2. Here the reviewer asked the authors to provide more solid evidence that, and how cigarette smoke induces a "reverse mode" of NCX1. Furthermore, the reviewer questioned the accuracy of the method used to examine intracellular calcium.

The authors have performed all experiments that were suggested by the reviewer. By performing classical calcium and

sodium imaging, they demonstrate that cigarette smoke, PMA and fMLP induce a very rapid and sustained increase in intracellular calcium. For cigarette smoke, this effect was strongly reduced in NCX1-knockout cells, and the NCX1-blocker KB-R7943 failed to inhibit effects in knockout cells. Cigarette smoke and PMA (but not fMLP) also induce a rapid reduction in intracellular sodium. With these data, the authors argue that their hypothesis about a reverse mode of NCX1 is strongly supported.

It is clear that the authors have performed all (or more) experiments that were required or suggested by reviewer 1. The new data seems to be conclusive, and it perfectly fits into the story told in the manuscript. As such, the authors made a great job in reacting to the comments of the reviewer.

My personal view on this is a little different. Essentially, the exact mechanism(s) accounting for cigarette smoke-induced modification of NCX1 remains poorly defined. Is it a direct effect on NCX1, or an indirect effect due to shifts of Calcium and Sodium over other mechanisms? Given that the authors only work on primary neutrophils, this question remains to be addressed. A straight forward approach to investigate this however, would be to examine recombinant NCX1 in an expression system allowing cellular imaging or electrophysiology specifically on NCX1.

A surprising, or even a little irritating observation in figure 7f, g and h, is that CSE, PMA and fMLP induce very rapid calcium and sodium fluxes. I believe that signals were recorded at ~0.3 Hz, thus all three substances induce effects that saturated within 3 seconds. Given that fMLP and PMA most likely do not directly modify NCX1, how can these very rapid effects by all three compounds be explained? Is it even plausible? Can artifacts from the application as such be ruled out?, i.e. a role of mechano-sensitive mechanisms like Piezo channels?

Cigarette smoke is a fairly well defined mixture of several substances. Which of these substances can induce this rapid effect? Again, experiments on NCX1 in simplified cellular system would enable the authors to address these points.

For now, the authors seem to consider these open but relevant questions as outside the scope of this study. Importantly, the authors do mention "Mechanistic understanding of NCX1 modes" as a point with limitations.

Version 2:

Reviewer comments:

Reviewer #2

(Remarks to the Author)

The authors have addressed all my concerns and provided informative new translational data to support their conclusions.

Reviewer #4

(Remarks to the Author)

The authors made a perfect job and addressed all my concerns with an astonishing accuracy.

I recommend publication.

A point-by-point response to referees' comments

REVIEWER COMMENTS

Reviewer #1 (Remarks to the Author):

The manuscript by Liao and coll. postulates that the reverse mode of NCX1 (*Slc8a1*) sodium/calcium exchanger is responsible for neutrophil-mediated lung damage in mice under conditions mimicking chronic obstructive pulmonary disease (COPD). Authors found enhanced expression of NCX1 in human samples from COPD individuals as well as from mice exposed to cigarette smoke. Authors propose that NCX1 acting in reverse mode, thus mediating calcium influx, is responsible for enhanced release of neutrophil extracellular traps (NETs). To support their claim, authors generate a conditional mouse model with selective ablation of NCX1 expression in neutrophils. This mouse model shows reduced formation of NETs upon stimulus and decreased COPD signs. However, the study lacks several control experiments) that are needed to support the conclusions, particularly regarding the direct involvement of NCX1-mediated calcium influx in promoting altered neutrophil function,

SPECIFIC POINTS

Figure 1 A. The specificity of the NCX1 antibody used for immunofluorescence in murine and human samples should be demonstrated by disappearance of the signal in neutrophils from KO animals.

Response: We thank the reviewer for raising this important point regarding antibody validation. To address this concern, we have performed additional immunofluorescence experiments using neutrophils isolated from our neutrophil-specific *Slc8a1* cKO mice (as described in the manuscript). Our immunofluorescence analysis clearly shows the absence of the NCX1 signal in neutrophils derived from *Slc8a1* cKO mice, compared to robust signal detection in control neutrophils. These new data are now shown as **Revised Supplementary Figure 3f** and incorporated into the revised manuscript (**lines 128-130, legends of Supplementary Fig. 3f**).

Supplementary Figure 2E. The expression of NCX1 detected by immunoblot in neutrophils from KO animals is markedly decreased but not absent. Is this a sign of a residual expression/function of the transporter?

Response: We appreciate the reviewer's keen observation regarding the residual NCX1 signal in immunoblots of *Slc8a1* cKO neutrophils. We agree that this warrants clarification. In our initial submission, we inadvertently omitted details regarding purity in the Methods section. Actually, as we mentioned in the Methods section, neutrophils were isolated by fluorescence-activated cell sorting (FACS) with 95%-98% purity (**lines 586-587**). Consequently, about 2%-5% of sorted cells are non-neutrophils (e.g., monocytes, macrophages, eosinophils), which express NCX1 endogenously. This small contaminating

population contributes detectably to the total protein lysate used for immunoblotting. We have included the post-sort purity FACS plots confirming this level of purity in the **Revised Supplementary Fig. 3c**. Besides, we have performed additional immunofluorescence experiments using neutrophils isolated from our neutrophil-specific *Slc8a1* cKO mice which shows the absence of the NCX1 signal in neutrophils derived from *Slc8a1* cKO mice (**Revised Supplementary Figure 3f**). We have updated the relevant content in figure panels accordingly (**lines 123-124, 129-130, legends of Supplementary Fig. 3f**).

In addition, our subsequent data also highlighted the substantially functional changes in the absence of NCX1 activity in *Slc8a1* cKO mice:

- 1) Significant depletion of NCX1 expression in *Slc8a1* cKO mice (**Revised Supplementary Fig. 3f**);
- 2) *Slc8a1* depletion showed significant protection against CS-induced NETs formation, lung damage and airway functions in COPD mice (**Revised Fig. 2**).
- 3) NETs formation in lung neutrophils was profoundly inhibited in CS-exposed *Slc8a1* cKO mice (**Revised Fig. 5f**);
- 4) Ca^{2+} influx was prominently depressed in *Slc8a1* cKO neutrophils (**Revised Fig. 7h**).

Collectively, our immunofluorescence and functional assays demonstrate that the genetic ablation effectively eliminates NCX1-mediated Ca^{2+} influx and its downstream pathological consequences (NETs formation and lung damage) in neutrophils. This confirms that the residual immunoblot signal does not affect the reliability of our experimental outcomes.

Figure 7. In my opinion, the claim that NCX1 is working in reversed mode and promoting high intracellular concentration is not adequately supported by experiments. In particular, the reverse mode of NCX1 is not an intrinsic characteristic of the transporter but depends on the ion gradients across the plasma membrane. An increase in intracellular sodium concentration can reverse the activity of NCX1 in neutrophils (Najder et al. Role of the intracellular sodium homeostasis in chemotaxis of activated murine neutrophils. *Front Immunol* 11:2124, 2020). Therefore, the effect of cigarette smoke may be indirect, increasing intracellular sodium. This issue has not been investigated. To study NCX1 function, authors use an unconventional method, based on a sensor that supposedly measures calcium flux direction. Looking at the vendor website, this instrument is mostly used for ion flux in plants. There are no tests in the paper to demonstrate that the method is really working compared to classical methods (e.g. measurement of intracellular calcium with ion-sensitive fluorescent probes). For example, there is no experiment showing that the calcium flux is abolished in neutrophils from NCX1 KO mice. Also, the NCX1 inhibitor should also be tested in neutrophils from KO animals to show that it does not alter flux when NCX1 expression/function is ablated. Additional experiments should have used methods to buffer or increase intracellular calcium by other means (e.g. calcium-chelating agents, calcium-mobilizing agonists, calcium ionophores) to check the effect on NET

release and/detection of calcium flux by the sensor.

Response: We sincerely thank the reviewer for these insightful critiques, which significantly strengthened our mechanistic investigation. We have now performed extensive new experiments to directly address these points.

1) Clarifying direct vs. indirect activation of reverse-mode NCX1 by stimuli

We fully agree that NCX1 reverse-mode activation depends on transmembrane ion gradients (e.g., elevated intracellular Na^+), commonly reported in references¹. To dissect whether cigarette smoke extract (CSE) acts on NCX1 via Na^+ dynamics, we simultaneously measured real-time changes in intracellular Na^+ ($[\text{Na}^+]_i$) and Ca^{2+} ($[\text{Ca}^{2+}]_i$) in neutrophils stimulated with three agents: CSE, PMA (phorbol ester), and fMLP (chemoattractant). Consistent with the reference mentioned by the reviewer¹, which reports indirect NCX1 reverse-mode activation secondary to Na^+ rise, our newly added data (Ca^{2+} measurement by Fura-2 AM; Na^{2+} measurement by NaTrium Green-2 AM) revealed that fMLP induced concurrent increases in both $[\text{Na}^+]_i$ and $[\text{Ca}^{2+}]_i$ (**Revised Fig. 7f-g**), confirming this indirect pattern. In stark contrast, CSE and PMA triggered rapid $[\text{Na}^+]_i$ decrease alongside robust $[\text{Ca}^{2+}]_i$ increase (**Revised Fig. 7f-g**). This divergence demonstrates CSE and PMA might act via a distinct, direct mechanism favoring reverse-mode NCX1 operation (Ca^{2+} influx/ Na^+ efflux), independent of prior mentioned $[\text{Na}^+]_i$ elevation (**lines 229-240**).

2) Addition of ion flux detection methodology and specificity validation of NCX1 inhibitor (KB-R7943) in neutrophils from *Slc8a1* cKO mice

To address concerns about the ion flux detection, we measured $[\text{Ca}^{2+}]_i$ with the standard fluorescent Ca^{2+} indicator Fura-2 AM, and $[\text{Na}^{2+}]_i$ measurement by NaTrium Green-2 AM. CSE and PMA stimulation triggered rapid $[\text{Na}^+]_i$ decrease alongside robust $[\text{Ca}^{2+}]_i$ increase (**Revised Fig. 7f-g**), and CSE-induced $[\text{Ca}^{2+}]_i$ rise was near-abolished in *Slc8a1* cKO neutrophils vs. Controls (**Revised Fig. 7h**), proving the critical role of NCX1 in CSE-triggered Ca^{2+} influx (**results: lines 229-240; methods: lines 650-664**). Furthermore, we also supplemented experiments to validate the specificity of NCX1 inhibitor on Ca^{2+} influx, which revealed that treatment with the NCX1 reverse-mode inhibitor KB-R7943 had no obvious effect on CSE-induced $[\text{Ca}^{2+}]_i$ in *Slc8a1* cKO neutrophils (**Revised Supplementary Fig. 4d, lines 248-251**), demonstrating its specificity and confirming successful NCX1 ablation.

3) Validation of NCX1-mediated Ca^{2+} influx and NETosis using Ca^{2+} ionophore and Ca^{2+} chelator

As requested, we applied Ca^{2+} ionophore (A-23187) and Ca^{2+} chelator (BAPTA-AM) to validate the causality between Ca^{2+} influx and NETs release. Pre-treatment with BAPTA-AM significantly reduced CSE-triggered $[\text{Ca}^{2+}]_i$ rise, Cit-H3 expression, and NETs formation (**Revised Fig. 7k, l; Revised Supplementary Fig. 4e, f; lines 245-257**). Neutrophils showed reduced protrusions consistent with inhibited NCX1 reverse mode activation. Meanwhile, treatment with A-23187 elevated $[\text{Ca}^{2+}]_i$, potently inducing Cit-H3 expression and NETs release (**Revised Supplementary Fig. 4e, f; lines 245-257**), confirming Ca^{2+} overload is key factor to drive NETosis. Furthermore, we conducted *in vitro* experiments using human peripheral blood neutrophils to validate that both the Ca^{2+} ionophore (A-23187) and the Ca^{2+} chelator (BAPTA-AM) modulate NETs formation, consistent with our findings in murine cells.

These results we provided in this revision demonstrate CSE directly engages reverse-mode NCX1 (evidenced by Na⁺ efflux/Ca²⁺ influx); validate the NCX1-dependent Ca²⁺ influx via Ca²⁺ measurement and genetic/pharmacologic controls; prove NCX1-mediated Ca²⁺ influx is necessary and sufficient for NETosis in this model. All new data have been incorporated into revised manuscript. We thank the reviewer for prompting these critical experiments, which substantially reinforce our mechanistic conclusions.

References

- 1 Najder, K. *et al.* Role of the Intracellular Sodium Homeostasis in Chemotaxis of Activated Murine Neutrophils. *Front Immunol* **11**, 2124, doi:10.3389/fimmu.2020.02124 (2020).

Reviewer #2 (Remarks to the Author):

Major strengths:

1. Fascinating findings supporting a novel mechanism in which the NCX1 transporter reverses its mode in neutrophils resulting in altered intracellular calcium and triggering NETosis.
2. The correlations in Figure 1I are very nice.
3. Beautiful SEM pictures throughout the manuscript.
4. The data shown in Figure 6 strengthened the manuscript.
5. Strengthened by both human and murine data.
6. Further made stronger by using multiple approaches to prove their mechanism (gene deletion, pharmacologic inhibition).

Response: We sincerely appreciate the reviewer's thoughtful and positive comments. We are glad to hear that our findings were considered fascinating and that the proposed mechanism regarding NCX1-mediated Ca^{2+} influx and its role in NETs formation was well received. We also appreciate the acknowledgment of key research design and outcomes in our manuscript. Furthermore, we are grateful for the recognition of the strength derived from using both human and murine data, as well as the multiple experimental approaches (gene deletion, pharmacological inhibition) to validate our hypothesis. In this revision, we have supplemented more data to make our conclusions more robust in accordance with your constructive suggestions.

Major critiques:

1. The nomenclature used of "simple emphysema" vs "COPD" is confusing and not consistent with clinical nomenclature as emphysema is a form of COPD. Would suggest using the term "mixed chronic bronchitis emphysema" vs "emphysema" to differentiate the two, if that is in fact what they mean.

Response: We sincerely appreciate the reviewer's professional suggestion regarding the use of clinically appropriate terminology. In the initial submission, we used the term "COPD" for simplicity and broader reader comprehension. However, we acknowledge that "emphysema" is indeed a subtype of COPD, and the terminology should be used precisely. Following your recommendation, we have revised the manuscript accordingly and now differentiate between "mixed chronic bronchitis and emphysema (mixed CBE)" and "emphysema" (**Revised Fig. 1b-k, Revised Fig. 5d-e, Revised Supplementary Fig. 1a-g; lines 87-113, 167-168**).

In addition, to enhance the clinical relevance of our findings, we further collected bronchoalveolar lavage fluid (BALF) and peripheral blood from five clinically diagnosed COPD patients in this revision. These patients had confirmed airflow limitation by pulmonary function tests and chest CT scans consistent with mixed CBE. Neutrophils isolated from these samples were used in functional assays, and in this context, we have adopted the term "mixed CBE" in accordance with your suggestion as well (**Revised Fig. 1l-o; lines 100-113**).

2. Figure 1C lists mRNA changes between "emphysema" and their "COPD" group, but the protein results for NCX1, MPO or NE in their "emphysema" group are not shown. Showing those results would strengthen their findings that the "simple emphysema" vs their "COPD/mixed chronic bronchitis emphysema" groups are molecularly distinct.

Response: Thank the reviewer for this excellent suggestion to strengthen our comparative analysis of the clinical phenotypes. As requested, we have now performed Western blot analyses of NCX1, MPO, and NE protein expression in samples from both emphysema group and Mixed CBE group. Protein levels of NCX1, MPO, and NE were significantly elevated in the Mixed CBE group compared to the emphysema group. This protein-level validation reinforces that neutrophil-associated pathology (NETosis drivers) is more pronounced in Mixed CBE. These new data are presented in **Revised Supplementary Fig. 1a-d** and described in the revised manuscript (**lines 92-97**).

Figure 1A: where in the lung are these tissue samples from? Hard to discern underlying lung structure. Perhaps labels of alveoli, etc would improve this.

Response: We appreciate the reviewer's request for histological clarity, and apologize for the lack of explanation in our original manuscript. The human lung sections in **Fig. 1a** were obtained from distal airways and bronchioles and adjacent alveolar structures, representing regions most affected in COPD pathogenesis. This panel primarily focused on assessing NCX1 co-localization with immune cells (lymphocytes, macrophages, dendritic cells and neutrophils). The observed characteristic nuclear distribution and tissue morphology (flattened epithelial nuclei lining airspaces) strongly suggest these regions primarily represent alveolar structures. To improve interpretability, we have updated the **Revised Fig. 1a** legend to explicitly state the tissue source, and supplemented more details in the revised manuscript (**lines 88, 838**).

4. Figure 1K: would be greatly strengthened if they could isolate neutrophils vs other immune cell subsets from patient BAL and run western blots for NCX1 expression. This may also be demonstrated by flow cytometry if NCX1 can be stained for by flow. The microscopy alone on lung tissue is insufficient to substantiate their statement that NCX1 is much higher in neutrophils vs other immune cells in control vs COPD patients.

Response: We sincerely thank the reviewer for this constructive suggestion. To directly address this important point, we have performed flow cytometry analysis using bronchoalveolar lavage fluid (BALF) samples from both control subjects (primarily asymptomatic pulmonary nodule patients) and COPD patients in this revision. Quantification of neutrophil count and NCX1 expression via flow cytometry revealed significantly greater number of neutrophils and higher NCX1 mean fluorescence intensity (MFI) in COPD neutrophils compared to controls (**Revised Fig. 1l-n**). We further confirm the neutrophil-specific NCX1 enrichment using flow cytometry, since NCX1 expression was markedly higher in neutrophils than in other immune cell populations (**Revised Supplementary Fig. 1e**). These results provide multiple lines of evidence that NCX1 is primarily upregulated in COPD neutrophils, and the increase is neutrophil-specific relative to other immune cells. These new data have been incorporated into Results, figure legends,

and Discussion sections, with corresponding methods updated (lines 100-113, 299-306, 581-613).

5. Are elevations in neutrophil NCX1 expression limited to lung neutrophils in COPD patients or is NCX1 expression increased in peripheral blood too?

Response: Thanks for raising this critical question about the systemic scope of NCX1 dysregulation. To address this, we isolated peripheral blood neutrophils using neutrophil cell extraction kit from healthy volunteers and COPD patients and performed Western blot analysis of NCX1 expression. As demonstrated, NCX1 protein levels were significantly elevated in peripheral blood neutrophils of COPD patients compared to healthy controls (**Revised Supplementary Fig. 1f, g**). This aligns with our lung tissue/BAL data, suggesting NCX1 dysregulation is systemic, not confined to pulmonary neutrophils, which may contribute to extrapulmonary COPD comorbidities. These new data have been included and discussed in the revised manuscript (lines 109-113, 299-303, 449-451, 586-587).

6. Demonstrate that *Slc8a1* deletion was neutrophil specific by isolating other immune cell types (mono/macros, etc) from these animals and showing NCX1 protein remains present in similar concentrations to control animals.

Response: Thanks for underscoring the importance of cell-type specificity in our genetic model. To validate the neutrophil-specific deletion of *Slc8a1* (NCX1), we isolated and analyzed non-neutrophil immune cells: flow-sorted non-neutrophils (monocytes/macrophages (Mono/Macs), lymphocytes, and dendritic cells) from both Control and neutrophil-specific *Slc8a1* cKO mice. Quantification of NCX1 expression via RT-qPCR (mRNA) found no significant difference in non-neutrophil immune cells between *Slc8a1* cKO and Control groups (**Revised Supplementary Fig. 3d**). Western blot (protein) exhibited comparable NCX1 protein expression in non-neutrophil immune subsets between *Slc8a1* cKO and Control groups (**Revised Supplementary Fig. 3e**). In contrast, neutrophils from *Slc8a1* cKO mice showed near-complete loss of NCX1 at both mRNA and protein levels (**Revised Supplementary Fig. 3d, e**). Immunofluorescence further validated the absence of NCX1 signal exclusively in neutrophils of *Slc8a1* KO mice (**Revised Supplementary Fig. 3f**). These data unequivocally demonstrate that *Slc8a1* ablation is primarily restricted to neutrophils. The results are now integrated into the revised manuscript with more details updated (lines 123-130).

7. Figures 3/4: While demonstrating that neutrophil numbers in each compartment are similar in both genotypes and showing that neutrophil function of bone marrow neutrophils are similar in both genotypes are solid starts, particularly as it relates to #5 above, would suggest testing functions of lung neutrophils between both genotypes as well.

Response: Thank the reviewer for this insightful suggestion, which has strengthened our investigation of compartment-specific neutrophil dysfunction in COPD. As recommended, we performed functional analyses of lung-infiltrating neutrophils in Control and neutrophil-specific *Slc8a1* cKO mice exposed to CS. CS-exposed *Slc8a1* cKO mice showed significantly reduced Cit-H3 levels in mouse BALF and NETs area in lungs

compared to controls (**Revised Fig. 5f-h**). No obvious differences were observed in Cit-H3 levels between Control and *Slc8a1* cKO mice under normal air exposure. Additionally, we detected levels of several chemokines and cytokines secreted by neutrophils, which revealed significantly diminished release of pro-inflammatory mediators in CS-exposed cKO mice relative to controls, including TNF- α , IL-8, IL-6, IL-1 β , and IL-17a (**Revised Fig. 5i, l-p**). This suggest that the critical role of NCX1 in neutrophil-mediated immune microenvironment. Moreover, CS-triggered pulmonary dysfunction and COPD pathological changes exhibited improvement in *Slc8a1* cKO mice (**Revised Fig. 2**). Together, these new data demonstrate that NCX1 deletion specifically improves pathogenic functions of lung neutrophils (pulmonary function, Cit-H3 levels, NETs area, cytokine release, COPD pathological changes). The results are now integrated into the revised maunscript, with more details updated in Methods section (**lines 164-189, 131-147, 324-371, 495-519**).

8. Figure 5: In addition to microscopy, would be much stronger if use an additional method to quantify NETs to substantiate the authors conclusions. For example, could measure NE: DNA, MPO:DNA or CitH3: DNA complexes by duel-ELISAs or Cit-H3 ELISA in the murine BAL. Same critique applies to Figure 7F & H.

Response: We sincerely appreciate the reviewer's suggestion to strengthen our NETs quantification through additional methods. To address this concern, we have performed ELISA assays of Cit-H3 to quantify NETs formation across key figures in this revision. *In vitro*, we isolated bone marrow neutrophils from mice (mice with two genotypes; WT mice), and treated the cell culture supernatants with CSE or PMA stimulation. Cit-H3 levels were significantly decreased either with NCX1 depletion or inhibition of NCX1 reverse mode (KB-R7943 treatment) (**Revised Fig. 5c-h, Revised Fig. 7i, Revised Fig. 7l, Supplementary Fig. 4c**). *In vivo*, we collected BALF from air or CS-exposed mice (*Slc8a1* cKO vs. Control), and found reduced Cit-H3 levels in CS-induced *Slc8a1* cKO mice (**Revised Fig. 5h**). Besides, Cit-H3 expression in BALF from WT mice with/without Cl-amidine (NETs inhibition) or KB-R7943 (NCX1 reverse mode inhibition) treatment have also been detected, which exhibited prominent reduction in mice with NETs formation suppressed (**Revised Fig. 6e**), or reverse mode of NCX1 inhibited (**Revised Fig. 8e**). These new data have been integrated into Results, Discussion sections, and corresponding methods have been supplemented in the Methods section (**lines 164-171, 241-248, 203-208, 266-271**).

9. Insufficient evidence to substantiate the proposed mechanism re NETs impeding the normal regression of tissue resident neutrophils from the lung, described in lines #191-193. There are several potential explanations including NETs promoting the release of chemokines. This has been previously demonstrated in lung epithelium by the authors of PMID # 36703990 and in fibroblasts by the authors of PMID # 23536012. Could measure the chemokine and cytokine levels in the BAL and/or whole lung to substantiate this as a mechanism.

Response: We sincerely appreciate the reviewer's insightful suggestion regarding the mechanistic link between NETs and neutrophil retention in the lung. As recommended, we have now performed chemokine/cytokine (TNF- α , IL-8, IL-6, IL-1 β , and IL-17a) profiling

in lung tissues from Control and neutrophil-specific *Slc8a1* cKO mice exposed to CS. Our newly added data revealed that CS exposure triggered significant upregulation of pro-inflammatory mediators in Control neutrophils. Whereas *Slc8a1* cKO mice exhibited markedly reduced levels of all measured cytokines/chemokines relative to Controls (**Revised Fig.5I-p**). This highlights that NCX1 help sustain neutrophil accumulation and retention to create a chemokine-rich microenvironment. Given prior reports that NETs amplify chemokine production in lung epithelium² and fibroblasts³, we speculate NCX1-driven NETs formation is a key upstream regulator of chemokine-rich microenvironment favoring neutrophil retention. These new data substantiate our proposed mechanism by highlighting a self-amplifying loop (NCX1--chemokine release--neutrophil retention). NCX1 inhibition could break this cycle by reducing NETs release and downstream chemokine production. These data are presented and discussed in the revised manuscript (**lines 172-189, 341-371**), with mentioned references cited in Discussion section (**line 356**). We thank the reviewer for prompting this critical validation, which significantly strengthens the mechanistic depth of our study.

References

- 2 Hudock, K. M. *et al.* Alpha-1 antitrypsin limits neutrophil extracellular trap disruption of airway epithelial barrier function. *Front Immunol* **13**, 1023553, doi:10.3389/fimmu.2022.1023553 (2022).
- 3 Khandpur, R. *et al.* NETs are a source of citrullinated autoantigens and stimulate inflammatory responses in rheumatoid arthritis. *Sci Transl Med* **5**, 178ra140, doi:10.1126/scitranslmed.3005580 (2013).

10. Figure 6: Need to confirm the Cl-amidine treatment decreased NET generation as they expect by microscopy or ELISA, etc.

Response: Thanks for this important suggestion to validate the effects of Cl-amidine on NETs formation. As requested, ELISA assay was performed to detect Cit-H3 levels in the BALF from CS-exposed mice. The results demonstrated that Cl-amidine treatment significantly reduced Cit-H3 levels in CS-exposed mice compared to control mice without treatment (**Revised Fig. 6e**). In addition, we supplemented immunostaining of NE/MPO/Cit-H3 in mouse lung tissues and found that decreased NETs area in Cl-amidine-treated mice (**Revised Fig. 6g, h**). These results have been incorporated into the revised manuscript (**lines 203-208, 506-519, 542-568**).

11. For Figure 7I: are there microscopy pictures that coincide with the graphs enumerating changes in the lungs?

Response: Thank the reviewer for requesting this important visual validation. In the revised manuscript, representative microscopy images corresponding to the mean linear intercept (MLI) quantification in Fig. 7i have been included as **Revised Fig. 8h, j, l, n**. These histological analysis of lung sections clearly demonstrate enhanced-alveolar

destruction and airway inflammation in CS-exposed mice, while these were reversed in the CS-exposed mice with inhibition of NCX1 reverse mode. These images and related description are now integrated in the revised manuscript (**lines 271-280**). We agree this addition strengthens the pathological evidence supporting our conclusions about NCX1's role in COPD progression.

12. The authors use PMA to induce NETs for comparison. While PMA is a robust stimulator of NETs, would suggest they also compare to more physiologically relevant stimuli such as LPS or IL-8 (Figure 2,5 and 7).

Response: We appreciate the reviewer's valuable suggestion regarding additional use of more physiologically relevant NETs stimuli. Actually, we compared the effects of several stimuli—including LPS, fMLP, and Cigarette Smoke Extract (CSE)—on NCX1 expression in neutrophils. Among these, CSE most robustly upregulated both mRNA and protein levels of NCX1 (**Revised Fig. 7a–d**). Given that cigarette smoke is the dominant risk factor for COPD, we prioritized Cigarette Smoke Extract (CSE) as our main stimulus to maximize clinical relevance. We also found that CSE stimulation elicited a comparable increase in Cit-H3 expression to that induced by PMA, the widely used positive control (**Revised Fig. 5c, Revised Fig. 7i, Revised Fig. 7l, Supplementary Fig. 4c**). In the cell culture supernatants of bone marrow neutrophils from mice, with either CSE or PMA stimulation, Cit-H3 levels were significantly decreased with NCX1 depletion or inhibition of NCX1 reverse mode (**Revised Fig. 5c, Revised Fig. 7i, Revised Fig. 7l, Revised Supplementary Fig. 4c**). This consists with our *in vivo* outcomes that CS exposure led to increased Cit-H3 levels and expanded NETs area in mouse BALF and lung tissue (**Revised Fig. 5f–h, Revised Fig. 6e–g, Revised Fig. 8e–g**). These data have been updated in the revised manuscript (**lines 220-223, 241-248**).

13. Likely, outside the scope of the current study, however, showing that the *Slc8a1* deleted mice have improvement in their pulmonary function tests after CSE would be an outstanding demonstration of the functional impact of NCX1 transporter in a COPD model.

Response: Thank the reviewer for this exceptional suggestion to strengthen the translational impact of our findings. We recognized the importance of pulmonary function tests and have now performed comprehensive respiratory physiology measurements. Our new results revealed striking improvements in CS-exposed *Slc8a1* cKO mice, indicated by enhanced airway function (increased FEV_{20(ms)}/FVC ratio and FEV_{50(ms)}/FVC ratio) and reduced air trapping (decreased FRC/TLC ratio) compared to Controls (**Revised Fig. 2b–d**). These data suggested preserved expiratory flow rates at critical timepoints and improved lung volume regulation in *Slc8a1* deleted COPD mice. Similar improvements were seen with NETs blockade (CI-amidine) (**Revised Fig. 6b–d**) and pharmacological inhibition of NCX1 reverse mode (KB-R7943) (**Revised Fig. 8b–d**) *in vivo*. We also supplemented air controls in comparison with CSE induction, demonstrating no obvious differences of pulmonary functions between *Slc8a1* cKO and Control mice under air exposure, but substantial dysfunctions in CS-exposed groups (**Revised Fig. 2b–d**). These results provide the direct evidence that NCX1 ablation and NETs formation blockade could restore airway functions of COPD mice. These data have been integrated into the Results, Methods, discussion of the revised manuscript (**lines 133-139, 196-202, 261-266**).

14. Discussion needs to be strengthened by a deeper discussion of how these findings are consistent or inconsistent with prior studies, providing enhanced context for the current findings and how the results truly advance the field.

Response: We sincerely appreciate this constructive suggestion. We have now significantly expanded the Discussion section to provide deeper integration with prior literature and clarify the conceptual advances of our study. The revised discussion has been strengthened with our newly added empirical data and more than 20 new references added. We believe these changes better position our work within the field's continuum while highlighting its transformative potential (**lines 282-422**).

Minor critiques:

1. Abstract: several typos and minor grammatical errors need correcting

Response: Thank the reviewer for pointing this out. We have thoroughly revised the Abstract to correct all typographical and grammatical errors. All corrections were marked in red in the revised manuscript (**lines 28-41**).

2. Abstract: write out what MPO and NE are abbreviations for

Response: Thank you for the reviewer's kind reminder. In the revised manuscript, we have modified the Abstract and there's no mention on these two words now due to content adjustment. We now use their full names at first mention—myeloperoxidase (MPO) and neutrophil elastase (NE) in the Results section (**line 96**). We have ensured consistency in abbreviation usage throughout the manuscript.

3. This statement as it currently reads, is incorrect, lines #63-64: "...COPD patients may suffer from drug induced neutropenia..." That is not a known lab finding in COPD unless they are referring to result of a specific therapy that is not widely used? Also a reference is needed for this statement.

Response: We have revised the sentence accordingly to improve its accuracy and clarity, and we have added an appropriate reference to support the revised statement. The updated sentence now reads "However, neutrophil-targeted therapies remain challenging in COPD due to the risk of drug-induced neutropenia and subsequent infection.", and the citation has been included in the revised manuscript (**lines 53-55**).

4. Figure 1A legend or the text: indicate what immune cell types that markers listed in green are meant to identify, e.g. CD3= T cells. Of note, both CD66b and CD11c are present on neutrophils.

Response: Thank the reviewer for this helpful suggestion. We have revised the legend of Figure 1A and the relevant text to indicate the immune cell types identified by the labeled markers. Thank you for your reminder on both CD66b and CD11c present on neutrophils. To address this concern, we performed triple immunostaining of NCX1⁺CD66b⁺/CD11c⁺ to further robustly identify neutrophils and NCX1 expression in these cells. Indeed, we observed major NCX1 co-localization with CD66b⁺/CD11c⁺ cells, highlighting the predominant expression of NCX1 in neutrophils. These data have been incorporated into the revised manuscript (**lines 89-90, 836-839**).

5. In section 2.2, indicate that the *Slc8a1* is the gene for NCX1.

Response: We have now indicated in Introduction section that *Slc8a1* is the gene encoding NCX1 (Na²⁺-Ca²⁺ Exchanger 1) to provide clarity for readers (**line 65**).

6. Clarify if NET area is measured where there is overlap of at least two relevant markers

Response: Thank the reviewer for this important methodological clarification. We confirm that NETs area quantification was performed using co-localization criteria with NET-specific markers (MPO, Cit-H3) to ensure accurate identification. Our stringent multi-marker approach minimizes false positives and aligns with current NETs research standards⁴. All NETs measurements required simultaneous positivity for Cit-H3 and MPO. Representative images are shown in **Revised Fig. 5d** (human lung tissues), **Fig. 5f** (mouse lung tissues), **Revised Fig. 6g** (Cl-amidine treatment), **Fig. 8f** (KB-R7943 treatment). Methodological and figure legend details have been added in the revised manuscript (**lines 567-568, 916-921, 942-945, 986-988**). The reviewer's suggestion helped us improve transparency in NETs identification criteria.

References

4 Zifkos, K. et al. Endothelial PTP1B Deletion Promotes VWF Exocytosis and Venous Thromboinflammation. *Circ Res* 134, e93-e111, doi:10.1161/circresaha.124.324214 (2024).

7. Section 2.4: Note Cl-amidine is a pan-PAD inhibitor and is not specific for PAD4.

Response: We appreciate the reviewer's accurate observation regarding Cl-amidine's pharmacological properties. As correctly noted by the reviewer, Cl-amidine inhibits multiple PAD isoforms (PAD1-4). We sincerely apologize for any confusion caused by our initial wording. Actually, in this part, we aimed to test the effects of NETs blockade on COPD pathology, without intent to dissect individual PAD isoform contributions. Thus, Cl-amidine was employed as a NETs inhibitor rather than a PAD4-specific probe. This aligns with its established use for broad NETs suppression^{5,6}. Our data also consistently demonstrated Cl-amidine treatment contributed to marked Cit-H3 decline and NETs area decreased, along with subsequent attenuation of pulmonary function and COPD pathology. This confirms the successful blockade of NETs release by Cl-amidine treatment. We apologize for the inappropriate description about this inhibitor in the original manuscript, which has been corrected in the Results and Methods sections of the revised manuscript (**lines 190-196**).

References

5 Shen, Y., You, Q., Wu, Y. & Wu, J. Inhibition of PAD4-mediated NET formation by cl-amidine prevents diabetes development in nonobese diabetic mice. *Eur J Pharmacol* **916**, 174623, doi:10.1016/j.ejphar.2021.174623 (2022).

6 Shen, Y., You, Q., Wu, Y. & Wu, J. Inhibition of PAD4-mediated NET formation by cl-amidine prevents diabetes development in nonobese diabetic mice. *European Journal of Pharmacology* **916**, 174623, doi:<https://doi.org/10.1016/j.ejphar.2021.174623> (2022).

Reviewer #3 (Remarks to the Author):

In the manuscript of Liao et al., the investigators characterized the role of NCX1 in neutrophils and its contribution to the pathogenesis of COPD.

They utilized in vitro and in vivo models including human samples showing significant upregulation of NCX11 in neutrophils, which correlated with MPO and NE. Furthermore, they showed attenuation of the neutrophil response in COPD including decreased emphysema in smoke-exposed mice with a neutrophil specific NCX1 deletion. Furthermore, cigarette smoke led to NCX1 to operate in reverse transport mode causing increased intracellular calcium and abnormal NET formation. Inhibition of these processes also attenuated CS-induced chronic bronchitis and development of emphysema like changes.

Overall, this manuscript describes the involvement of neutrophil NCX1 in the COPD lung in addition to elucidation the downstream signaling pathways that are mediating the lung changes and neutrophil response thereby describing a potential novel pathway to target in the neutrophilic COPD phenotype. There are some moderate concerns that should be addressed.

1) Supplemental figure 1A will need quantification of airway disease (MLI etc).

Response: We thank the reviewer for this suggestion to strengthen our supplemental data. We have now performed comprehensive quantitative analysis of airway and parenchymal pathology in Supplemental Figure 1A (**Revised Supplementary Fig. 2a, b**). The quantifications of mean linear intercept confirm that mice with CS exposure for 6 and 12 weeks exhibited alveolar destruction. These data have been integrated into the revised Results section, and highlighted in the updated figure legends (**lines 115-118, legends of Revised Supplementary Fig. 2a, b**).

2) Fig. 2D-J, control mice are missing, that are not exposed to cigarette smoke.

Response: Thank the reviewer for highlighting this important omission. We have now included comprehensive data from air-exposed control groups in pulmonary function tests (FEV_{20(ms)}/FVC, FEV_{50(ms)}/FVC, FRC/TLC), HE staining for MLI and collagen deposition quantification, α -SMA immunofluorescence, Masson's trichrome, and SFTPC+ cell immunofluorescence (**Revised Fig. 2**). No significant differences were found between two groups under air exposure, but CS-exposed mice exhibited prominently significant pulmonary dysfunctions and pathological changes compared to air-exposed control groups. The inclusion of air-exposed controls confirms that NCX1 deletion does not alter baseline lung structure and strengthens the injury-specific interpretation of our KO phenotype. We have updated all relevant methodological details, Results and figure legends in the revised manuscript (**lines 114-147, 865-869**).

3) Fig. 3 says that there was no difference in neutrophil number, but Fig. 2C shows the difference in number in the lung – this should be explained/it is not made clear whether these mice were exposed to cigarette smoke in Fig 3 (if not, it would be hard to measure migration to the lung if not stimulated). This is definitely a group that should be included.

Response: We appreciate the reviewer's careful attention to this important distinction. We regret any misunderstanding due to our insufficient clarification in the initial submission. In this revision, we have clarified the experimental design and added critical missing data to resolve any confusion. Fig. 2C (**Revised Fig. 2f**) showed significantly increased neutrophil number in CS-exposed Control mice (12-week exposure) compared to CS-exposed *Slc8a1* cKO mice. Meanwhile, we analyzed air-exposed mice and confirmed no difference in neutrophil numbers between Control and *Slc8a1* cKO groups (**Revised Fig. 2f**). Data analyzed in Fig. 3 were all from non-CS exposed mice, since this section aimed to investigate the contribution of NCX1 in neutrophil function. We concluded that *Slc8a1* (NCX1) deletion reduced NETs formation of neutrophils, without affecting development, maturation, migration and phagocytosis of neutrophils. More details were provided in the Results and Figure legends of the revised manuscript (**lines 139-147, 150-152, 865-872, 896-897**).

4) Fig. 6 G does not correlate with the images and should be shown with better representative images.

Response: Thank the reviewer for this important observation regarding image representation. We have carefully addressed this concern by replacing the original Fig. 6G (**Revised Fig. 6k**) images with new, more representative Masson's trichrome staining ones. The updated images now clearly correspond to the quantitative data in **Fig. 6l**.

5) The PAD4 inhibitor should be better described and should be validated. Neutrophils are not the only cells that express PAD4 and this should be taken into consideration.

Response: We appreciate the reviewer's accurate observation regarding Cl-amidine's pharmacological properties. As correctly noted by the reviewer, Cl-amidine inhibits multiple PAD isoforms (PAD1-4). We sincerely apologize for any confusion caused by our initial wording. Actually, in this part, we aimed to test the effects of NETs blockade on COPD pathology, without intent to dissect individual PAD isoform contributions. Thus, Cl-amidine was employed as a NETs inhibitor rather than a PAD4-specific probe. This aligns with its established use for broad NETs suppression^{5,6}. Our data also consistently demonstrated Cl-amidine treatment contributed to marked Cit-H3 decline and NETs area decreased, along with subsequent attenuation of pulmonary function and COPD pathology. This confirms the successful blockade of NETs release by Cl-amidine treatment. We apologize for the inappropriate description about this inhibitor in the original manuscript, which has been corrected in the Results and Methods sections of the revised manuscript (**lines 190-196**).

References

- 5 Shen, Y., You, Q., Wu, Y. & Wu, J. Inhibition of PAD4-mediated NET formation by cl-amidine prevents diabetes development in nonobese diabetic mice. *Eur J Pharmacol* **916**, 174623, doi:10.1016/j.ejphar.2021.174623 (2022).
- 6 Shen, Y., You, Q., Wu, Y. & Wu, J. Inhibition of PAD4-mediated NET formation by cl-amidine prevents diabetes development in nonobese diabetic mice. *European Journal of Pharmacology* **916**, 174623,

doi:<https://doi.org/10.1016/j.ejphar.2021.174623> (2022).

6) FlexiVent analysis of these mice would be helpful.

Response: Thank the reviewer for this exceptional suggestion to strengthen the translational impact of our findings. We recognized the importance of pulmonary function tests and have now measure obstructive ventilation dysfunction (FEV_{20(ms)}/FVC ratio and FEV_{50(ms)}/FVC ratio) and residual capacity (FRC/TLC) by Buxco/DSI system (DSI BuxcoPFT, America). Our new results revealed striking pulmonary function improvements in CS-exposed *Slc8a1* cKO mice, indicated by increased FEV_{20(ms)}/FVC ratio and FEV_{50(ms)}/FVC ratio and decreased FRC/TLC ratio compared to Controls (**Revised Fig. 2b-d**). Similar improvements were demonstrated with NETs blockade (CI-amidine) (**Revised Fig. 6b-d**) and pharmacological inhibition of NCX1 reverse mode (KB-R7943) (**Revised Fig. 8b-d**) *in vivo*. We also supplemented air controls in comparison with CSE induction, demonstrating no obvious differences of pulmonary functions between *Slc8a1* cKO and Control mice under air exposure, but substantial dysfunctions in CS-exposed groups (**Revised Fig. 2b-d**). These results provide the direct evidence that NCX1 ablation and NETs formation blockade could restore airway functions of COPD mice. These data have been integrated into the Results, Methods, discussion of the revised manuscript (**lines 133-139, 196-203, 261-266**).

7) Please comment on how many mice (male, female) were used and how many total donors and replicates in the Methods Statistics section.

Response: Thank the reviewer for emphasizing the importance of transparent experimental reporting. We have now included detailed information about sample used and replicates of mice and human donors in both the Methods and figure legends (**lines 834-1011**). All original data including biological and technical replicates were provided in the **Source data** file uploaded with resubmission.

Minor:

1) The PAD4 inhibitor and specificity for NETs should be introduced and validated.

Response: We appreciate the reviewer's accurate observation regarding CI-amidine's pharmacological properties. As correctly noted by the reviewer, CI-amidine inhibits multiple PAD isoforms (PAD1-4). We sincerely apologize for any confusion caused by our initial wording. Actually, in this part, we aimed to test the effects of NETs blockade on COPD pathology, without intent to dissect individual PAD isoform contributions. Thus, CI-amidine was employed as a NETs inhibitor rather than a PAD4-specific probe. This aligns with its established use for broad NETs suppression^{5,6}. Our data also consistently demonstrated CI-amidine treatment contributed to marked Cit-H3 decline and NETs area decreased, along with subsequent attenuation of pulmonary function and COPD pathology. This confirms the successful blockade of NETs release by CI-amidine treatment. We apologize for the inappropriate description about this inhibitor in the original manuscript,

which has been corrected in the Results and Methods sections of the revised manuscript (lines 190-196).

References

- 5 Shen, Y., You, Q., Wu, Y. & Wu, J. Inhibition of PAD4-mediated NET formation by cl-amidine prevents diabetes development in nonobese diabetic mice. *Eur J Pharmacol* **916**, 174623, doi:10.1016/j.ejphar.2021.174623 (2022).
- 6 Shen, Y., You, Q., Wu, Y. & Wu, J. Inhibition of PAD4-mediated NET formation by cl-amidine prevents diabetes development in nonobese diabetic mice. *European Journal of Pharmacology* **916**, 174623, doi:<https://doi.org/10.1016/j.ejphar.2021.174623> (2022).

2) The neutrophil isolation should be described in more detail: in the methods section, it is said that they were isolated from BALB/c mice, but the models used are all C57BL/6

Response: We sincerely apologize for this oversight in our original submission. All mouse studies were indeed performed using C57BL/6 mice. The erroneous reference to BALB/c has been modified throughout the manuscript, and more methodological details about animal strain and neutrophil isolation have been updated in Methods section (lines 472-476, 582). We deeply regret this error and have implemented additional verification steps in our manuscript preparation process to prevent such issues in future submissions. The corrections have been highlighted in the revised manuscript with track changes for easy identification.

3) Were neutrophils isolated via flowcytometry sorting?

Response: We appreciate this opportunity to clarify our neutrophil isolation methodologies. Our approach differed by sample type to ensure optimal purity and viability. Human blood neutrophils were isolated using negative selection kits (EasySep™ Human Neutrophil Enrichment Kit, StemCell #19257). Murine neutrophils from bone marrow were isolated via FACS sorting (gating strategy: CD45⁺CD11b⁺Ly6g⁺), with purity >95% (**Revised Supplementary Fig. 3c**). We have updated Methods section and all relevant figure legends to specify isolation methods (lines 581-613).

References

- 1 Najder, K. *et al.* Role of the Intracellular Sodium Homeostasis in Chemotaxis of Activated Murine Neutrophils. *Front Immunol* **11**, 2124, doi:10.3389/fimmu.2020.02124 (2020).
- 2 Hudock, K. M. *et al.* Alpha-1 antitrypsin limits neutrophil extracellular trap disruption of airway epithelial barrier function. *Front Immunol* **13**, 1023553, doi:10.3389/fimmu.2022.1023553 (2022).
- 3 Khandpur, R. *et al.* NETs are a source of citrullinated autoantigens and stimulate inflammatory responses in rheumatoid arthritis. *Sci Transl Med* **5**, 178ra140, doi:10.1126/scitranslmed.3005580 (2013).
- 4 Zifkos, K. *et al.* Endothelial PTP1B Deletion Promotes VWF Exocytosis and Venous Thromboinflammation. *Circ Res* **134**, e93-e111, doi:10.1161/circresaha.124.324214 (2024).
- 5 Shen, Y., You, Q., Wu, Y. & Wu, J. Inhibition of PAD4-mediated NET formation by cl-amidine prevents diabetes development in nonobese diabetic mice. *Eur J Pharmacol* **916**, 174623, doi:10.1016/j.ejphar.2021.174623 (2022).
- 6 Shen, Y., You, Q., Wu, Y. & Wu, J. Inhibition of PAD4-mediated NET formation by cl-amidine prevents diabetes development in nonobese diabetic mice. *European Journal of Pharmacology* **916**, 174623, doi:<https://doi.org/10.1016/j.ejphar.2021.174623> (2022).

REVIEWER COMMENTS

Reviewer #2 (Remarks to the Author):

Authors provide a thoughtful and thorough response addressing all of my critiques and concerns.

One important clarification needs to be made: The authors refer to measuring IL-8 in mice in Figure 5, but mice do not express IL-8, which is a bit concerning. The murine homologues are CXCL-1, CXCL-2 and CXCL-5 and are important neutrophil chemoattractants. What exactly did the authors measure? Did they use murine primers or human primers for this RT-PCR?

Response:

We sincerely thank the reviewer for this important reminder. After checking on the primer design website, we confirmed that, the primers labeled as “IL-8” in our RT-PCR experiments using tissues from the conditional knockout mice were actually designed to target CXCL-2, the murine homologue of human IL-8. We apologize for this oversight and for not clearly distinguishing between human and mouse nomenclature in the manuscript. We have now corrected the gene name accordingly throughout the text and figure panels in the revised version (**Revised Fig. 5m, lines 190, 656, 987**).

Revised Fig. 5m RT-qPCR detected mRNA levels of TNF- α , CXCL-2, IL-6, IL-1 β , and IL-17a in mouse lungs with different genotypes.

I would strongly consider showing pressure-volume loops which they should be able to generate and would be visually more clear as the FEV1(20) and (50) are not used commonly used clinically. The authors likely also have compliance and elastance data too that would strengthen their findings as well of emphysema or lack thereof.

Response:

Thank you very much for your valuable and constructive suggestions. We fully agree with your recommendation to include pressure-volume loops, as they provide a clearer and more intuitive visualization of the data, and serve as a more effective representation of pulmonary function. Upon further discussion with our technical team, we realized that pressure-volume recordings were indeed part of the data we had previously collected. Thanks to your insightful suggestion, we reanalyzed and visualized the complete pressure-volume loops in the revised manuscript (**Revised Fig. 2b, Revised Fig. 6b, Revised Fig. 8b**). Additionally, we do have compliance data as well, which have been incorporated into the revised manuscript (**Revised Fig. 2c, Revised Fig. 6c,**

Revised Fig. 8c). Pressure–volume loops and compliance data demonstrated that deletion of NCX1 significantly ameliorated CS-induced decrease of lung compliance (**Revised Fig. 2c**). Consistently, PAD4 inhibition by Cl-amidine (**Revised Fig. 6c**) and pharmacological inhibition of NCX1 reverse-mode activity with KB-R7943 (**Revised Fig. 8c**) improved dynamic compliance (C_{dyn}). These additional data enhance the robustness of our interpretation that NCX1 reverse-mode activity aggravates lung mechanical dysfunction in COPD. We have now incorporated the newly added data into the text and figure panels in the revised manuscript (**lines 133-140, 202-206, 285-288, 536-542, 929-932, 998-1002, 1046-1050**).

Revised Fig. 2b-c The pressure-volume curves and dynamic compliance data of *Slc8a1* cKO mice and Control mice under normal air or CS exposure.

Revised Fig. 6b-c The pressure-volume curves and dynamic compliance data of Control, CS, CS+Vehicle and CS+Cl-amidine mice.

Revised Fig. 8b-c The pressure-volume curves and dynamic compliance data of Control, KB-R7943, CS, CS+KB-R7943 mice.

Reviewer #3 (Remarks to the Author):

All concerns have been appropriately addressed.

Response:

Thank you very much for your positive evaluation. We sincerely appreciate your careful review and are grateful for your recognition that all concerns have been appropriately addressed.

Reviewer #4 (Remarks to the Author):

Dear ladies and gentlemen,

I was asked to re-review new data and comments given to address the concerns of reviewer #1 following the initial submission of the manuscript.

Reviewer #1 only had three specific points, and the authors have included several new data to address those:

Point 1 - Specificity of the NCX1 antibody - The authors now very clearly demonstrate the specificity of the NCX1 antibody, the new data are very convincing.

Point 2 - Remaining expression of NCX1 in neutrophils from knockout animals - The authors now demonstrate that the purity of neutrophils following FACS is "only" 95%-98%. Thus, 2-5% of the cells are other immune cells that still express NCX1. This is convincing and sufficient to address this concern.

Point 3 - This point is more "major" than point 1 and 2. Here the reviewer asked the authors to provide more solid evidence that, and how cigarette smoke induces a "reverse mode" of NCX1. Furthermore, the reviewer questioned the accuracy of the method used to examine intracellular calcium.

The authors have performed all experiments that were suggested by the reviewer. By performing classical calcium and sodium imaging, they demonstrate that cigarette smoke, PMA and fMLP induce a very rapid and sustained increase in intracellular calcium. For cigarette smoke, this effect was strongly reduced in NCX1-knockout cells, and the NCX1-blocker KB-R7943 failed to inhibit effects in knockout cells. Cigarette smoke and PMA (but not fMLP) also induce a rapid reduction in intracellular sodium. With these data, the authors argue that their hypothesis about a reverse mode of NCX1 is strongly supported.

It is clear that the authors have performed all (or more) experiments that were required or suggested by reviewer 1. The new data seems to be conclusive, and it perfectly fits into the story told in the manuscript. As such, the authors made a great job in reacting to the comments of the reviewer.

My personal view on this is a little different. Essentially, the exact mechanism(s) accounting for cigarette smoke-induced modification of NCX1 remains poorly defined. Is it a direct effect on NCX1, or an indirect effect due to shifts of Calcium and Sodium over other mechanisms? Given that the authors only work on primary neutrophils, this question remains to be addressed. A straight forward approach to investigate this however, would be to examine recombinant NCX1 in an expression system allowing cellular imaging or electrophysiology specifically on NCX1.

Response: Thank you for this important suggestion. To directly address whether cigarette smoke-related stimuli act on NCX1 directly rather than indirectly, we employed CHO-K1 cells in which NCX1 was not present based on literature reports^{1,2},

which were also confirmed by our western blot and immunofluorescence data (**Revised Supplementary Fig. 4h, i**). After CHO-K1 cells transfected with recombinant NCX1, the real-time $[Ca^{2+}]_i$ and $[Na^+]_i$ were monitored using ion-sensitive fluorescent probes. Compared with vector controls, NCX1-overexpressing CHO-K1 cells displayed a rapid rise in intracellular Ca^{2+} accompanied by a decrease in intracellular Na^+ upon stimulation (**Revised Supplementary Fig. 4j, k**), consistent with our results demonstrating Ca^{2+}/Na^+ fluxes stimulated by CSE in primary neutrophils. As you've concerned about mechanosensitive artifacts in the next question below, this experiment included pre-incubation with GsMTx4 to block Piezo channels. The CHO-K1 overexpression data, together with NCX1 depletion in Slc8a1 Cko and pharmacological inhibition of NCX1 reverse mode (KB-R7943), make a compelling case that NCX1 mediates the CSE-triggered NCX1 reverse-mode Ca^{2+} influx we observed in primary neutrophils. The related text and figure panels have been updated in the revised manuscript (**lines 257-263, 418-420, 686-697, 712-717, legends of Supplementary Fig. 4**).

Revised Supplementary Fig. 4h, i Western blot analysis and immunofluorescence validated the expression of NCX1 in NCX1-transfected CHO-K1 cells and vector controls.

Revised Supplementary Fig. 4j, k Ca^{2+} and Na^{2+} influx in NCX1-transfected CHO-K1 cells and vector controls (pre-incubated with GsMTx4) upon CSE stimulation

A surprising, or even a little irritating observation in figure 7f, g and h, is that CSE, PMA and fMLP induce very rapid calcium and sodium fluxes. I believe that signals were recorded at ~0.3 Hz, thus all three substances induce effects that saturated within

3 seconds. Given that fMLP and PMA most likely do not directly modify NCX1, how can these very rapid effects by all three compounds be explained? Is it even plausible? Can artifacts from the application as such be ruled out?, i.e. a role of mechano-sensitive mechanisms like Piezo channels?

Response:

Thank you for your thoughtful observation. We fully acknowledge your concern regarding the rapid calcium and sodium fluxes observed in Figure 7f, g, and h following treatment with CSE, PMA, and fMLP, which may be caused by the mechanical stimulation during the drug application through activating mechanosensitive channels, leading to the transient ion fluxes that you have pointed out.

To address this concern, we performed additional cellular experiments using GsMTx4, a specific inhibitor of Piezo channels³, to block mechanosensitive ion channels before addition of different stimuli. The results of these experiments revealed after inhibiting Piezo channels, the time required for Ca²⁺ and Na⁺ fluxes to reach their peak was delayed compared to the untreated condition, while the trends of peak fluxes themselves remained unchanged in later several minutes (**Revised Supplementary Fig. 4d-f**). Notably, after the initial delay, the ion fluxes stabilized and continued at a consistent level. In our response to your first question, Piezo inhibition (GsMTx4 pretreatment) was also performed in detecting Ca²⁺ and Na⁺ fluxes in NCX1-overexpressing CHO-K1 cells with CSE stimulation, revealing swift Ca²⁺ elevation and Na⁺ decrease in NCX1-transfected CHO-K1 cells (**Revised Supplementary Fig. 4j, k**). These findings strongly suggest that mechanosensitive Piezo channels indeed influence the initial speed of Ca²⁺ and Na⁺ fluxes, but do not affect the sustained flux direction and the overall stability of the ion fluxes over time.

We believe this newly-added experiment successfully rules out mechanical stimulation as a significant artifact affecting the sustained calcium and sodium fluxes, as Piezo inhibition did not alter the long-term stability of the ion fluxes. This provides further confidence that CSE, PMA, and fMLP triggered NCX1 activity rather than mechanical artifacts of Piezo channels.

Once again, we sincerely appreciate your thoughtful comments, which have prompted us to conduct these additional experiments. We believe the revised manuscript with results and related details updated (**lines 250-263, 686-697, 712-717, legends of Supplementary Fig. 4**), now provides a more rigorous and scientific explanation for the observed phenomena.

Revised Supplementary Fig. 4d-f Simultaneous real-time measurements of intracellular Ca²⁺ (F_{340/380}) and Na⁺ (F_{500/545}) in GsMTx4 pre-incubated neutrophils by fluorescent probes upon stimulation.

Revised Supplementary Fig. 4j, k Ca²⁺ and Na⁺ influx in NCX1-transfected CHO-K1 cells and vector controls (pre-incubated with GsMTx4) upon CSE stimulation.

Cigarette smoke is a fairly well defined mixture of several substances. Which of these substances can induce this rapid effect? Again, experiments on NCX1 in simplified cellular system would enable the authors to address these points.

Response: We greatly appreciate the reviewer’s insightful question. Cigarette smoke (CS) contains thousands of chemicals—including oxidants, reactive aldehydes, nicotine, particulate matter, and various free radicals. The U.S. Food and Drug Administration established a list of 93 harmful and potentially harmful constituents in tobacco products⁴, highlighting the extreme complexity of CS composition. Given the vast number of components, verifying each component and their collaborative relationships would be a massive research undertaking.

Our study focused on investigating the key effector cell type (neutrophils) associated disease mechanisms in a physiologically relevant setting—CS-induced COPD. The whole-smoke exposure might better mimic the authentic pathological environment. Nevertheless, we fully agree that determining which specific toxicants may trigger rapid NCX1-related responses is an important future direction. Studies using simplified cellular systems and selected CS constituents (e.g., acrolein, ROS donors, nicotine, or particulate extracts) would indeed help clarify the direct molecular triggers. We plan to incorporate such mechanistic analyses in our future work to further elucidate the component-specific effects. This has been mentioned as limitations in our Discussion section (**lines 447-448**). Thanks again for your valuable suggestions on our work.

For now, the authors seem to consider these open but relevant questions as outside the scope of this study. Importantly, the authors do mention "Mechanistic understanding of NCX1 modes" as a point with limitations.

Response: We fully understand the reviewer’s concerns. We acknowledge that the mechanistic questions related to NCX1 modes are important and biologically relevant. Within the current revision, we have made substantial efforts to strengthen the scientific rigor by adding multiple complementary experiments and analysis, including pressure–volume analysis, dynamic compliance data, ion fluxes monitoring with mechanosensitive channels depressed, and validation in recombinant NCX1 cellular expression system. These additions reinforce the robustness of our conclusions, and provide a more comprehensive picture of how NCX1 reverse mode contributes to

COPD pathogenesis. Nevertheless, this study did not investigate the molecular structural basis underlying the transition between NCX1's reverse and forward modes. Understanding the mechanisms driving this mode switching remains an important open question in the field. We acknowledge that fully dissecting the molecular determinants that drive NCX1 transition between forward and reverse modes—especially in response to complex CS stimuli—requires integrating more high-resolution structural biology with real-time functional analyses and dedicated mechanistic approaches. This point has been discussed in the limitations section of the revised manuscript (**lines 447-453**). Your suggestions have been invaluable in improving the quality and clarity of our manuscript. We are greatly thankful for your time and effort in reviewing our work.

References

- 1 Wan, H. *et al.* NCX1 coupled with TRPC1 to promote gastric cancer via Ca(2+)/AKT/ β -catenin pathway. *Oncogene* **41**, 4169-4182, doi:10.1038/s41388-022-02412-9 (2022).
- 2 Long, Y. *et al.* Functional comparison of the reverse mode of Na⁺/Ca²⁺ exchangers NCX1.1 and NCX1.5 expressed in CHO cells. *Acta Pharmacol Sin* **34**, 691-698, doi:10.1038/aps.2013.4 (2013).
- 3 Rothman, A., Wolner, B., Button, D. & Taylor, P. Immediate-early gene expression in response to hypertrophic and proliferative stimuli in pulmonary arterial smooth muscle cells. *J Biol Chem* **269**, 6399-6404 (1994).
- 4 Cheng, T., Reilly, S. M., Feng, C., Walters, M. J. & Holman, M. R. Harmful and Potentially Harmful Constituents in the Filler and Smoke of Tobacco-Containing Tobacco Products. *ACS Omega* **7**, 25537-25554, doi:10.1021/acsomega.2c02646 (2022).

REVIEWERS' COMMENTS

Reviewer #2 (Remarks to the Author):

The authors have addressed all my concerns and provided informative new translational data to support their conclusions.

Reviewer #4 (Remarks to the Author):

The authors made a perfect job and addressed all my concerns with an astonishing accuracy. I recommend publication.

Response: We would like to express our sincere gratitude to Reviewer #2 and Reviewer #4 for their thoughtful and constructive comments. We greatly appreciate the time and effort they dedicated to reviewing our manuscript. Their feedback was instrumental in improving the quality of our study, and we are pleased to hear that the revisions we made addressed their concerns. We are also grateful for their positive evaluation and recommendation for publication.

Thank you again for your invaluable input.